# Direct interaction of a chaperone-bound type III secretion substrate with the export gate

Dominic Gilzer [1], Madeleine Schreiner[1] & Hartmut H. Niemann [1✉]

Several gram-negative bacteria employ type III secretion systems (T3SS) to inject effector proteins into eukaryotic host cells directly from the bacterial cytoplasm. The export gate SctV (YscV in *Yersinia*) binds substrate:chaperone complexes such as YscX:YscY, which are essential for formation of a functional T3SS. Here, we present structures of the YscX:YscY complex alone and bound to nonameric YscV. YscX binds its chaperone YscY at two distinct sites, resembling the heterotrimeric complex of the T3SS needle subunit with its chaperone and co-chaperone. In the ternary complex the YscX N-terminus, which mediates YscX secretion, occupies a binding site within one YscV that is also used by flagellar chaperones, suggesting the interaction's importance for substrate recognition. The YscX C-terminus inserts between protomers of the YscV ring where the stalk protein binds to couple YscV to the T3SS ATPase. This primary YscV–YscX interaction is essential for the formation of a secretion-competent T3SS.

[1] Department of Chemistry, Bielefeld University, Universitaetstrasse 25, 33615 Bielefeld, Germany. ✉email: hartmut.niemann@uni-bielefeld.de

Type III secretion systems (T3SSs; also: injectisomes, virulence-associated vT3SS) are a common device of bacterial virulence that is present in gram-negative bacteria such as *Salmonella*, *Pseudomonas*, or *Yersinia*[1]. The macromolecular complex is composed of approximately 20 different protein species, many of which are present in multiple copies, and spans both bacterial membranes. Injectisomes share high structural similarity and likely a common ancestor with the secretion system found in bacterial flagella (fT3SS). With the needle-like character of purified injectisomes[2] and their capability to secrete effector proteins directly into a host cell, the T3SS can also be described as a 'molecular syringe' and may be a useful protein delivery tool[3].

Secretion through the *Yersinia* T3SS is strictly regulated by the inner membrane-anchored export apparatus containing YscR-STUV and can be divided into three distinct phases: (i) 'early' substrates, which self-assemble into the extracellular needle; (ii) 'intermediate' substrates, hydrophobic translocator proteins that insert themselves into the host membrane, thereby forming a direct connection between bacterial and host cytosols; and (iii) 'late' substrates or effector proteins that are then directly funneled through the injectisome into the host cytoplasm[4]. Many substrates of the T3SS are transported to the export apparatus in complex with a type III secretion chaperone (T3SC), which keeps the substrate in a partially unfolded and secretion-competent conformation. The ATPase complex then separates the complex under ATP-hydrolysis[5] and allows for the secretion of the substrate in an unfolded or partially helical state[6].

T3SCs can further be subdivided into three classes. Class I chaperones bind effector proteins before injection. Class II binds to hydrophobic translocator proteins. Class III is a heterogeneous group and contains chaperones of flagellin in fT3SS or the injectisomal needle subunit SctF (YscF). Class II chaperones contain three tetratricopeptide repeat (TPR) motifs[7] where one TPR is made up of two antiparallel α-helices. The tandem arrangement of TPRs generates a convex outer side that forms extensive interactions with the translocator and a concave binding groove where a short stretch of the substrate molecule binds in an extended conformation[8–11]. Additionally, a gatekeeper protein is involved to ensure timely secretion of translocators before effector proteins[12]. For the class III chaperone of the needle component SctF, two structures (YscEFG from *Yersinia* and PscEFG from *Pseudomonas*) have been published[13,14]. Both show heterotrimeric complexes of SctF with the two chaperones YscG/PscG and YscE/PscE, where SctF binds into the concave groove of the TPR protein YscG/PscG via a short α-helix. Interestingly, the second chaperone YscE/PscE, which mainly consists of two α-helices, caps the N-terminus of YscG/PscG but barely interacts with the substrate.

Injectisomes can be divided into seven families[15], such as the Inv-Mxi-Spa family present in *Salmonella enterica*, *Shigella* sp. or *Escherichia coli*, or the Ysc family associated with *Yersinia* sp., *Pseudomonas aeruginosa*, *Aeromonas* sp. or *Photorhabdus luminescens*. Ysc family T3SS produce two proteins, SctX and SctY (YscX and YscY in *Yersinia*), that are not found in other T3SS families. Functionally, little is known about these two proteins except that they are required for the export of early substrates and hence the assembly of a functional T3SS[16–18]. YscX and its *Aeromonas* homolog AscX are secreted with YscY acting as chaperone for YscX prior to secretion[17,19]. YscX is not an early substrate, since its secretion occurs after the YscP-mediated substrate specificity switch[18]. It has been shown that the N-terminus of YscX harbors its secretion signal, while its C-terminus is essential for the formation of a secretion-competent T3SS[17]. While no experimental structures are available for either protein, YscY can be assigned a role as T3SC based

on its sequence which predicts 3 TPRs[20]. The structure of YscX remains elusive, given that its only homologs are other uncharacterized SctX proteins. The binding mode appears to be conserved between different SctX and SctY, as heterologous complexes from different species have been observed via yeast two-hybrid assays[21]. After separation from YscX, YscY probably serves a regulatory function within the bacterial cell due to its interaction with SycD[22,23], the chaperone escorting the hydrophobic translocator proteins YopB and YopD.

The interaction between YscX, YscY, and the major export apparatus protein YscV has been established to be a prerequisite for the timely secretion of early substrates and the tripartite complex SctVXY is conserved between different T3SS-carrying species[18,21]. SctVs belong to the FlhA superfamily and as such are structurally similar to their flagellar FlhA counterparts. The export gate may receive substrates previously bound by a cytosolic sorting platform[24,25]. Over the years, several structures of FlhA and SctVs have been published (FlhA:[26–28] SctV:[29–34]), recently amended by the electron microscopy (EM) structure of *Yersinia* YscV as a homo-nonamer[35]. In general, SctVs contain an N-terminal transmembrane domain followed by a linker and a large cytosolic domain. It has been shown that SctVs form oligomeric structures in vivo that are consistent with the nonameric ring observed in different crystal and EM structures[36]. The cytosolic domain can further be subdivided into four subdomains (SD1–SD4), where SD2 is an insertion in SD1. SD3 and SD4 are necessary for the formation of the nonameric ring[30,37]. NMR data of a ternary complex composed of the cytosolic domain of *Salmonella* FlhA, the substrate FliC and its chaperone FliS has demonstrated that for a flagellar T3SS, recognition of the substrate:chaperone complex is facilitated via the chaperone and a conserved region in SD2 of FlhA[38]. It is noted, however, that the recognition site is not conserved in injectisomal SctVs and deletion of the entire SD2 did not abolish secretion of translocator and tip proteins for *Shigella* MxiA[30]. Therefore, the binding of substrate:chaperone complexes might occur at a different position in virulence-associated T3SS than in fT3SS.

Here, we report the structure of the YscX:YscY complex alone, and as a tripartite complex with the cytosolic domain of YscV. Our results demonstrate that the binding mode of this injectisomal substrate:chaperone complex is distinct from the binding of flagellar chaperones to the export gate and they explain why the C-terminus of YscX is essential for its function as injectisomal component.

## Results

**Structure determination of the YscX:YscY complex**. To structurally characterize the two proteins YscX and YscY, we first attempted to express them independently as YscX fused to maltose-binding protein (MBP-YscX) and His$_6$-YscY in *E. coli*. While YscY was expressed insolubly, MBP-YscX was observed in the soluble fraction but showed heavy precipitation after cleaving off the solubility-promoting MBP-tag. We, therefore, tried co-expression of the two constructs, which yielded soluble protein. Each protein could pull down the other during affinity chromatography and YscY and YscX co-eluted in a symmetrical peak when subjected to size exclusion chromatography. Since stored samples of the complex showed clear signs of proteolytic digestion of YscX, limited proteolysis coupled with N-terminal sequencing and mass spectrometric analysis was conducted to identify a more stable N-terminally truncated version of YscX (YscX$_{32}$). Neither the C-terminal region of YscX nor YscY were sensitive towards proteolysis. A study by Day and Plano[17] further suggested that only the region YscX$_{50–110}$ is necessary for it to bind YscY, so further constructs truncating the N-terminus

(YscX$_{50}$), the C-terminus (YscX$_{32-110}$), or both (YscX$_{50-110}$) were created.

Our crystallization constructs contained N-terminally truncated YscX$_{50}$ and full-length YscY with an N-terminal hexahistidine tag. The complex crystallized in space group $P2_12_12$ with the asymmetric unit (AU) containing two heterodimers related via translational non-crystallographic symmetry. We solved the structure by molecular replacement (MR) using a model generated by AlphaFold 2 (Supplementary Fig. 1). Data collection and refinement statistics are reported in Supplementary Table 1. The two dimers in the AU are very similar with an r.m.s.d. value of 0.744 Å between the Cα residues of the complex. Differences lie mostly within YscX—where the r.m.s.d. between the two molecules is 0.904 Å—caused by a slight shift in the second helix (α2) of the protein.

**YscX and YscY form an entwined heterodimer**. The AlphaFold 2 model correctly predicted the overall fold of the complex with YscX binding to YscY at two distinct sites connected via a short linker, resulting in YscX entwining its chaperone (Fig. 1a). Similar to other T3SCs the concave face of YscY forms an interaction site to which YscX binds via a short helix α1 (S53–A61). The area of interaction for this site is approximately 500 Å². Moreover, this groove is largely lined by hydrophobic residues and allows for the binding of YscX's L57, W58, F60, A61, and P63 into distinct binding pockets. Hydrophilic residues such as K54, R55, and D59 point toward the solvent (Fig. 1b). S53 and S56 of YscX are 3.8 and 3.9 Å apart from Y101 and Y104 of YscY, respectively, making their contribution to the binding interface via hydrogen bonds unlikely. The importance of the amphipathic character of the binding helix in YscX is further underlined by the high degree of sequence conservation for the hydrophobic residues that interact with the chaperone when compared to the otherwise low

conservation of helix α1 (Fig. 2a). The linker between the binding helix and the C-terminal helices of YscX is only resolved in one of the two heterodimers in the AU, indicating the flexibility of the loop.

A second interface is formed via the interaction of YscY's N-terminal TPR and the two longer C-terminal α-helices (α2 and α3) of YscX. The helices of YscX are rotated by about 90° relative to YscY with the binding site containing the hydrophobic residues I3, L5, F12, L13, I31, L32, A35, and L39 for YscY as well as L80, L83, L86, L87, L95, A98, L101 and L102 for YscX (Fig. 1c). The total area of interaction averages out at 1436.7 Å² for both YscXY heterodimers (PISA[39]).

The C-terminal helix of YscX extends about 26 Å further out from where it last interacts with its chaperone, as measured from L102 to H120. A hydrogen bond of Q106 to D81 is the last intramolecular interaction between α2 and α3 of YscX. Therefore, the remaining residues (about 20 Å of the C-terminal helix) are not forming interactions within the YscX:YscY heterodimer. In the crystal, this helix is stabilized by interacting with helix α3 of the neighboring molecule as well as the same helix from symmetry-related molecules, forming a 4-helix interface. B-factors for the protruding part of helix α3 are relatively high in chain B, interestingly however not in chain D despite its generally higher B-factors, especially in the poorly defined loop and helix α2 (Supplementary Fig. 2). Furthermore, K121 and V122 are not ordered in the crystal, suggesting that the C-terminus of YscX is somewhat flexible.

**The YscX N-terminus is disordered**. In addition to our structure of YscX$_{50}$:YscY, we also crystallized a longer construct containing residues 32–122 of YscX (YscX$_{32}$:YscY). Crystallization of this complex required the reductive methylation of primary amines[40]. The resolution was considerably poorer, with the protein

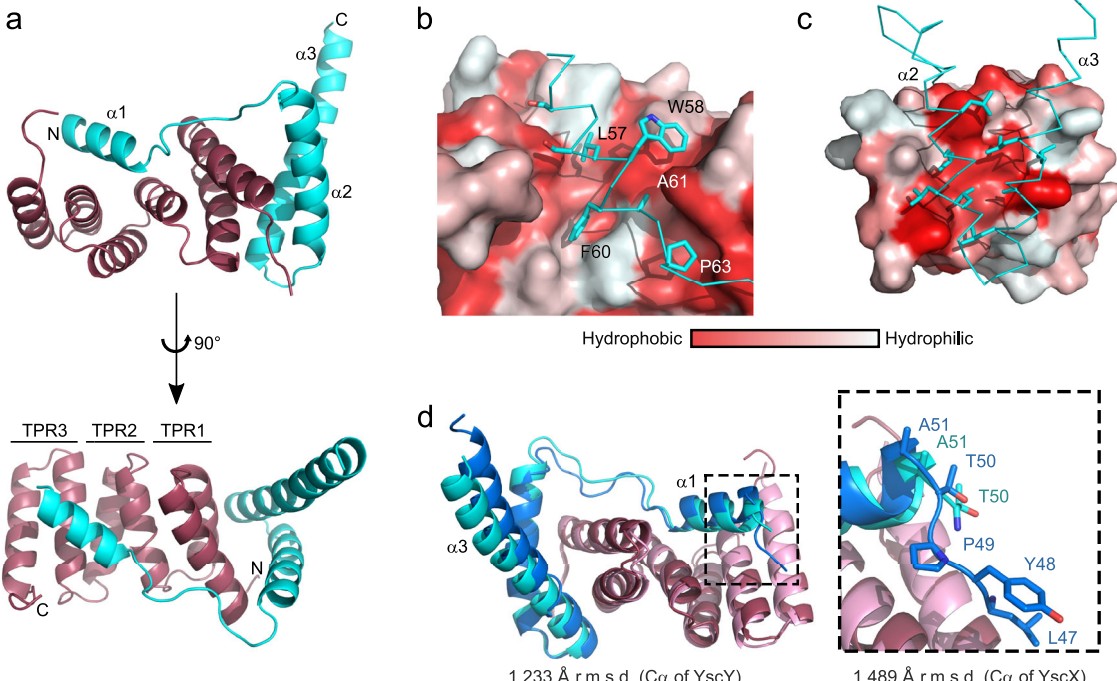

**Fig. 1 Overall structure of the YscX:YscY heterodimer. a** Cartoon representation of YscX (cyan) and YscY (red). YscX binds to YscY at two independent sites. **b, c** Surface hydrophobicity gradient of YscY shown from hydrophobic (red) to hydrophilic (white) using the normalized consensus hydrophobicity scale[65]. **b** YscX binds the groove of the TPR protein YscY via the hydrophobic side of a short amphipathic helix. **c** Two leucine-rich helices of YscX interact with the largely hydrophobic surface of the N-terminal TPR in YscY. **d** Superposition of YscX$_{50}$:YscY (cyan and red) and YscX$_{32}$:YscY (blue and pink) models. The alignment was calculated using the Cα atoms of YscY. Zoom-in shows the YscX N-terminus before α1 for both structures when aligned using the Cα atoms of YscX.

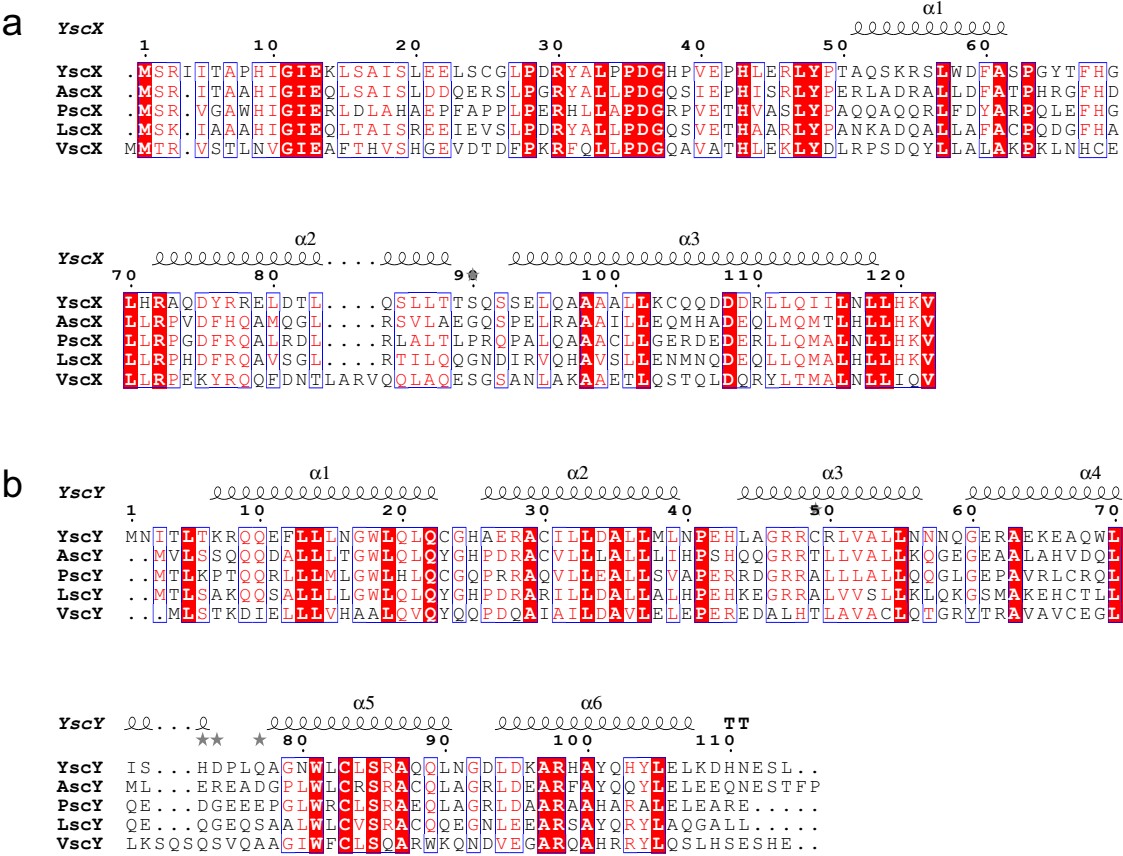

**Fig. 2 Sequence alignment of SctX and SctY. a** Alignment of SctX. **b** Alignment of SctY. Alignments were calculated using ClustalW[66] and visualized with ESPript3 (espript.ibcp.fr,[67]). SctX and SctY sequences from *Yersinia enterocolitica* (YscX: AAD16820.1, YscY: AAK69217.1), *Aeromonas hydrophila* (AscX: AAS91814.1, AscY: AAS91815.1), *Pseudomonas aeruginosa* (PscX: AAY17109.1, PscY: AAY17110.1), *Photorhabdus luminescens* (LscX: CAE16135.1, LscY: CAE16134.1), and *Vibrio parahaemolyticus* (VscX: UJX28051.1, VscY: EDM61220.1) were used to generate the alignment. Definition of colors and symbols: red box, white character: strict identity; red character: similarity in a group; blue frame: similarity across groups; gray star: residues with alternate conformations.

adopting a loose packing in space group $I4_122$ with a solvent content of 64.3% (Supplementary Table 1). The overall fold is identical to that of the shorter construct and the structures align well with r.m.s.d. values of 2.033 Å between YscX$_{32}$:YscY and chain A and B of YscX$_{50}$:YscY or 1.388 Å with chain C and D of the latter. This is compared to an r.m.s.d. of 0.744 Å between the two complexes in the AU of the YscX$_{50}$:YscY crystal. In general, the r.m.s.d. values were a bit higher for YscX (1.489 Å or 1.383 Å for chain B or chain D, respectively) than for YscY (1.233 Å or 1.078 Å) and the models differ the most in the position of the loop between helices α1 and α2 of YscX and in a slightly increased bending at the C-terminus of α3 in the model for YscX$_{32}$:YscY (Fig. 1d). Even though this YscX construct is 18 residues longer than YscX$_{50}$, only L47-P49 of these are resolved, suggesting that the N-terminus is largely disordered. Moreover, the C-terminus in this model is fully resolved and is helical albeit with high B-factors for the last residues (Supplementary Fig. 2).

**Structure determination of YscX:YscY bound to nonameric YscV.** We then attempted to co-purify the cytosolic domain of of YscV (YscV$_C$) with complexes of YscX and YscY from *E. coli*. Yields were considerably higher for the ternary complex (30 mg per liter of culture) than for the heterodimer (1–5 mg per liter). During gel filtration, the complex eluted as a species of high molecular weight. The assembly crystallized readily and conditions were identical regardless of which YscX construct (YscX$_{full-length}$, YscX$_{32}$, or YscX$_{50}$) was used. Diffraction measurement of the

ternary complex containing YscV$_C$, YscX$_{32}$, and His$_6$-YscY gave data to a resolution of 4.1 Å (Supplementray Table 1). We tested approximately 100 crystals of this complex with most not diffracting further than 8 Å if at all. The phase problem was solved via MR with an EM model of nonameric YscV (PDB: 7ALW). Two nonamers are stacked upside-down with their membrane-distal sides interacting in an assembly probably lacking biological significance also described by Kuhlen et al.[35] (Fig. 3a). We then attempted to place the model of YscX$_{50}$:YscY via MR as well, which failed most likely due to its relatively low scattering mass. Instead, 18 binary complexes were placed manually into helical density present outside of the nonameric YscV rings (Supplementary Fig. 3). The crystal described here is packed loosely with a solvent content of 67.2% that originates both from the central 5 nm-wide pore of the cyclic YscV$_C$ nonamer and the fact that the stacked nonamers hardly interact with one another, leaving a large volume of bulk solvent between the 18-mers (Supplementary Fig. 4). Instead, almost all crystal contacts occur between YscY molecules or between an YscY molecule and SD2 of YscV$_C$ which also points out from the ring.

**Overall structure of nonameric YscV$_C$.** In general, the electron density was reasonably well defined for subdomains 3 and 4 (SD3 and SD4), which are mostly buried within the ring. The average area of the YscV-YscV interaction is approximately 1600 Å$^2$ and is stabilized by four salt bridges between E431-R560, D511-R592, E517-R563, and R519-E531. The electron density of SD1 was

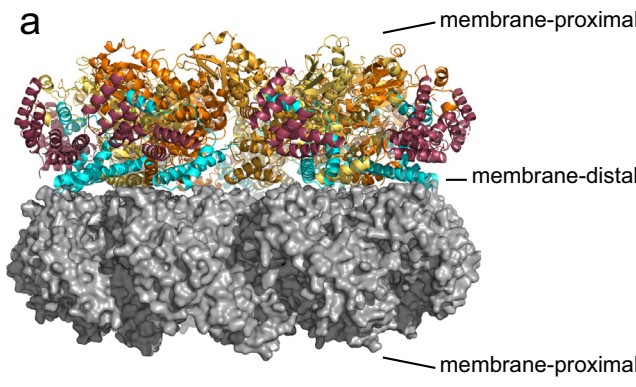

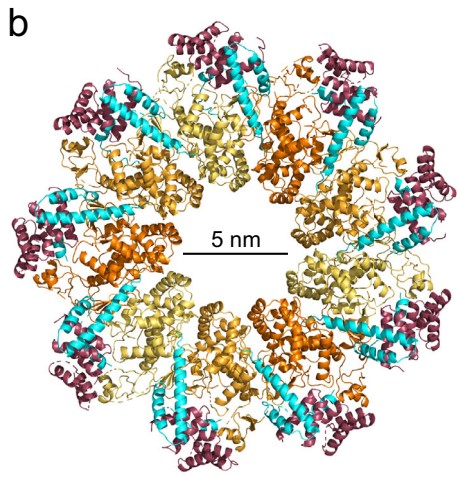

**Fig. 3 Overall structure of the ternary YscV$_C$:YscX$_{32}$:YscY complex.**
**a** Side View of the two stacked nonameric rings of YscV$_c$ (orange, three shades for clarity). Half of the chains are shown as gray surface. YscX$_{32}$ (cyan) and YscY (red) bind on the periphery of the nonamer. **b** View on the membrane-distal side of one nonamer. The C-terminal helix of YscX protrudes into the ring between two adjacent YscV$_C$ protomers.

interpretable, even though the central 4-stranded β-sheet was only visible as a continuous sheet of electron density. In contrast, SD2 showed the lowest map quality with broader variations between the protomers with regard to connectivity of the density. The cleft formed between SD2 and SD4 is in an open conformation, with a distance of roughly 10 Å between the two domains at the cleft's entrance. Parts of the linker between transmembrane and cytoplasmic domain are also visible (residues 359–371) and fold onto the neighboring protomer, where it runs along a defined groove, as previously observed[35]. We confirmed the sequence register of YscV$_c$ via anomalous difference density from a selenomethionine derivative (Supplementary Fig. 5).

The individual chains of the complex superimpose well (Supplementary Fig. 6a), especially the rigid SD3 of YscV$_C$. More flexibility can be observed for SD2 and at the outer regions of SD1 and SD4. YscX chains are very similar except for the linker between helices α1 and α2, which was poorly defined (Supplementary Fig. 6e). The binding of YscX:YscY to YscV$_C$ seems to allow some leeway, since an alignment of all substrate or all chaperone chains results in fanning out of the YscV$_C$ chains (Supplementary Fig. 6d−g).

**YscY and the YscX N-terminus bind the SD2-SD4 cleft.** Each YscX:YscY complex binds to the SD2-SD4 cleft of an individual YscV$_C$ protomer. The PISA server[39] reports interactions between

(A) YscY's first TPR and SD4, (B) the third TPR and SD2, as well as (C) a putative interaction between the extended N-terminus (L47 and Y48) of YscX with SD1 from inside the cleft between SD2 and SD4 (Fig. 4b–d). The latter interaction was only observed for chains FB, GB, KB as well as LB and may be biologically relevant since these residues are conserved between SctX. Due to the poor resolution, the exact nature of these interactions could not be determined as density was usually absent for side chains in YscX and YscY. For the aforementioned site (A) hydrophobic interactions could originate from I604, P606, L652, I655, P659, and L699 of YscV, and hydrogen bonds could arise between Y608, Q656, S657, and Q697 of YscV with Q22, H25, and R28 of YscY. Several leucines and isoleucines can be found in site (B), which may be complemented by polar interactions between Q452, E454, S486, and Q487 of SD2 as well as R86, Q89, R98, and Y101 of YscY. In case of site (C), the conserved Y48 of YscX might be interacting with aromatic residues of YscV in vicinity, such as F491 and W499 of SD1. Furthermore, Y48 is pointing towards a patch of different possible hydrogen bonding partners, such as R447, S449, D469, Q470, and E489 of YscV.

**The C-terminus of YscX binds between YscV protomers.** The C-terminal α3 helix—which is the extended helix seen in the YscX:YscY structures—is inserted between two YscV$_C$ protomers (Fig. 3b) at the cleft formed between two adjacent SD4 (hereafter referred to as SD4–SD4 cleft). This is the major binding interface between YscV and YscX and involves SD3 and SD4 of YscV. Except for a salt bridge formed between D108 of YscX and R701 of YscV$_C$ directly at the entry point for the helix, the interaction is largely hydrophobic (Fig. 5a) with most residues involved being highly conserved (Fig. 2a). The C-terminal helix of YscX does not extend all the way into the inner hole of the YscV nonamer but instead ends contacting the neighboring YscV molecule. The negative charge of the carboxy-terminus at V122 is presumably neutralized by R551 and R669 of a second YscV molecule (Figs. 4e and 5). Mutating R551 to alanine (R551A) or glutamate (R551E) reduced or abolished the ability of YscV$_C$ to be pulled down by MBP-YscX$_{50}$:His$_6$-YscY in MBP pull-down assays (Fig. 6a), indicating the importance of charge neutralization for recognition of YscX by the export gate.

**YscV requires the C-terminus of YscX for its recognition.** From the ternary complex structure, it became clear that the major mode of binding between the YscX:YscY heterodimer and the nonameric YscV is via an insertion of YscX's C-terminus between two YscV monomers within the ring (Figs. 3 and 4a) with further interactions involving the chaperone and SD2 or SD4 as well as residues L47 and Y48 of YscX. To test whether the inserting helix of YscX is necessary for binding to YscV, an MBP pull-down assay was employed, where MBP-YscX:YscY complexes were immobilized from lysate either with or without YscV$_C$ present. The SDS-PAGE (Fig. 6a) clearly demonstrated that the N-terminal truncation YscX$_{50}$ has no impact on binding, whereas deletion of the C-terminal 12 amino acids (YscX$_{32–110}$:YscY or YscX$_{50–110}$:YscY) abolished the interaction. This was underlined by analytical gel filtrations done with mixtures of either YscV$_C$ or a truncated version lacking SD3 and SD4 (YscV$_{SD12}$, residues 356–513)—which therefore does not contain the binding site for the C-terminal helix of YscX—with YscX32:YscY. Since YscX$_{32}$:YscY contains the putative binding region $^{46}$RLYP$^{49}$, it should be able to bind to SD1 via YscX and to SD2 via YscY.

On its own, YscX$_{32}$:YscY tends to produce a three-peak pattern during gel filtration, most likely due to the formation of oligomers in solution (Fig. 6b). At a concentration of 10 µM, YscV$_C$ forms a single peak at around 60 kDa apparent molecular mass. As

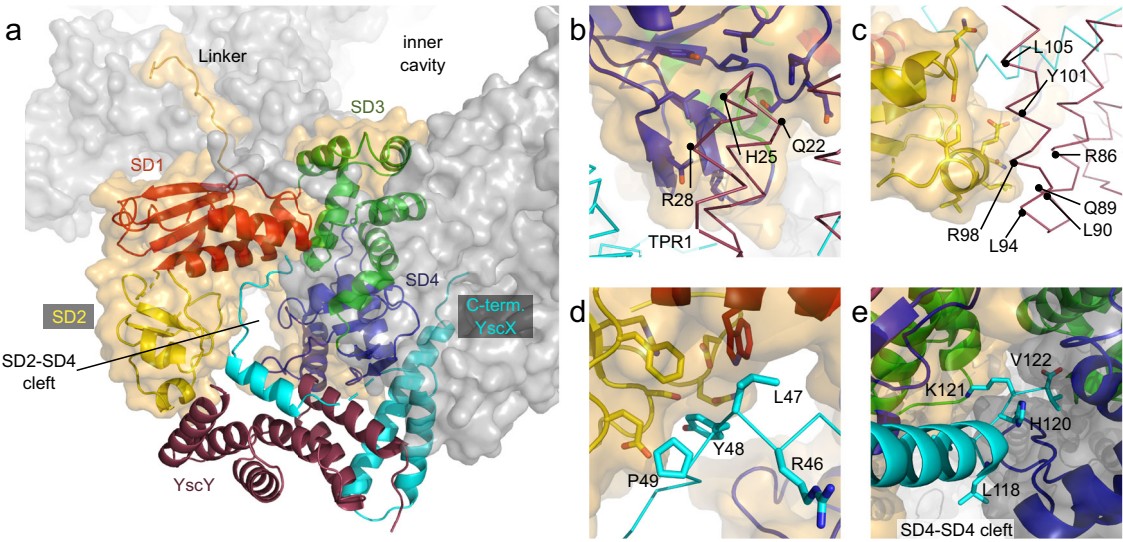

**Fig. 4 Interactions of YscV, YscX, and YscY.** One YscV$_C$ protomer is highlighted (orange surface) within the nonameric ring (gray surface). The cartoon representation of YscV$_C$ is divided into the four subdomains: SD1 (red, residues 372–438 and 493–513), SD2 (yellow, residues 439–492), SD3 (green, residues 514–600), and SD4 (blue, residues 601–704). For clarity, YscX (cyan) and YscY (dark red) are shown only for this heterotrimer and are depicted as ribbon in (**b**–**d**). **a** The overall structure of the heterotrimer within the assembly shows that all interactions, except for the insertion of YscX's C-terminus, are confined within one heterotrimer, i.e., YscY does not interact with a neighboring YscV subunit. **b** YscY interacts with SD4 via the loop of its first TPR (site A) and **c** with SD2 via the face of its C-terminal TPR (site B). **d** In chains FB, GB, KB, and LB, YscX exhibits density for an extended N-terminus which binds to SD1 and SD2 via a conserved $^{46}$RLYP$^{49}$ motif (site C). Note that the exact positions of the side chains could not be determined at this resolution and are only included for completeness. **e** The C-terminus of YscX is inserted into the SD4–SD4 cleft. Both involved YscV$_C$ subunits are shown as cartoon colored by subdomain. YscX is shown as ribbon with sidechains starting from L118.

discussed later, the shift from the expected weight of 40.5 kDa was attributed to dynamic assembly and disassembly of the nonameric ring during the run (Fig. 7). With a mixture of YscV$_C$ and the binary complex, a clear shift towards higher apparent molecular masses can be seen as a peak around 10 mL retention volume (Fig. 6b top, Supplementary Table 2, Supplementary Fig. 8), which we attributed to a nonameric ring of YscV$_C$ with an unknown quantity of YscX:YscY bound to it. For the ternary complex, two more peaks were observed at approximately 14 and 15 mL, the latter of which could be attributed to free YscV$_C$ in solution or a 1:1:1 ternary complex for which a molecular mass of 65 kDa would be expected. The large and broad signal at around 14 mL might also indicate other oligomeric forms and a dynamic equilibrium between different states. The shortened YscV$_{SD12}$ construct elutes after approximately 16 mL, which corresponds to around 36 kDa, potentially representing a dimer (Fig. 6b, bottom). Here, no shift in mass was observed when the protein was mixed with YscX$_{32}$:YscY in an equimolar fashion and thus we concluded that SD1 and SD2 alone do not bind YscX:YscY with high affinity. A construct of YscV$_C$ only containing SD3 and SD4 (residues 514–704) could not be analyzed as it was insoluble upon expression in *E. coli*.

**The termini of YscX oppositely influence nonamerization of YscV$_C$.** YscV forms higher oligomers in *Yersinia* regardless of whether YscX and YscY are present[36,37] and micelle-embedded YscV assembles into nonamers in vitro[35]. For the isolated cytosolic domain of other SctV proteins, both monomeric, as well as nonameric species, have been observed in solution[31,32,34]. With the major recognition site of YscX:YscY and YscV$_C$ being located at the far C-terminus of YscX and between two YscV protomers, we aimed at investigating a possible influence of YscX on the nonamerization of the export gate's cytosolic domain. Size exclusion chromatography of YscV$_C$ at different concentrations revealed a concentration-dependent dynamic equilibrium

between different states with the nonameric form being stabilized at higher concentrations (Fig. 7, top). At 2.5 μM initial concentration of YscV$_C$, the protein appears to be largely monomeric as indicated by a symmetrical peak at around the expected molecular mass of 40.5 kDa. At 10 or 25 μM, we observed a shift in the apparent mass and significant fronting, indicating the dynamic association and dissociation of higher oligomeric species on the column. At 100 μM the main peak is shifted to a high apparent molecular mass corresponding to a nonamer, but there is still notable tailing, i.e., incomplete conversion to the nonameric ring. We then compared the oligomerization behavior in the presence of two different YscX:YscY constructs, which were co-purified with YscV$_C$. At 10 μM, YscV$_C$X$_{50}$Y showed almost-complete conversion to higher masses. Interestingly, however, the full-length YscX-containing ternary complex (YscV$_C$X$_{fl}$Y) favored the monomeric species at the same molar concentration of protein (Fig. 7, middle and bottom).

Using the same concentration series as for YscV$_C$, the nonameric ring of YscV$_C$X$_{50}$Y is stable even at an initial concentration of 2.5 μM, whereas full-length YscX seems to inhibit ring formation even at high concentrations (Fig. 7, bottom). We concluded that the N-terminus of YscX is not necessary for binding YscV, but it decreases the propensity of YscV$_C$ to nonamerize and therefore must interact in some other way.

## Discussion

YscX and YscY are largely uncharacterized proteins from the T3SS of *Yersinia* with a yet unknown function despite being essential for a functional injectisome. With YscX belonging to a unique and elusive protein family without any homologous structures, we were unable to solve the phase problem via MR. After laborious attempts at experimental phasing, we attempted to predict the structure of the two proteins computationally using I-TASSER[41], tFold (to be published), RoseTTAfold[42], and

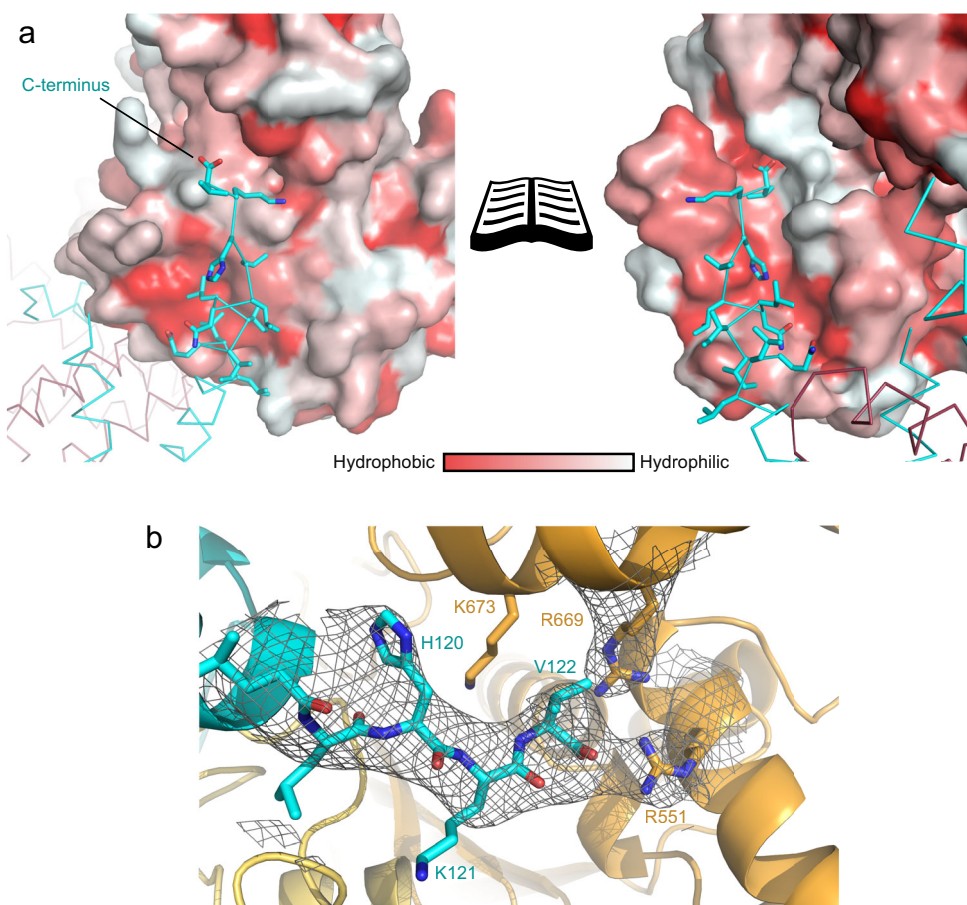

**Fig. 5 The C-terminus of YscX binds between two YscV protomers. a** Surface hydrophobicity of YscV (left) and the neighboring protomer (right) from hydrophobic (red) to hydrophilic (white) with a ribbon and stick representation of YscX (cyan). Helix α3 inserts between two YscV molecules and binds via hydrophobic interactions. The C-terminus is pointing towards a positively charged patch of the neighboring subunit, which likely stabilizes the carboxylate. **b** $2mF_O\text{-}DF_C$ density contoured at 1σ at the C-Terminus of YscX suggests positioning of the carboxylate towards R551 of YscV (orange). The density of the arginines is exemplary and varies between the chains.

AlphaFold 2[43]. YscY was always predicted as a TPR protein and the general fold was very similar between programs (Supplementary Fig. 1b). YscX on the other hand produced models that were only similar for the C-terminal α2 and α3 helices (Supplementary Fig. 1a). Finally, we attempted to predict the complex structure using the same programs but supplying both sequences fused together by a 33-residue linker. While the other predictions were disregarded, we were intrigued by how compatible the AlphaFold 2 model was to our biochemical and crystallographic observations. Indeed, we could solve the phase problem for our YscX:YscY crystals with a truncated version of the AlphaFold 2 complex model. In retrospect, models of YscY were also sufficient for MR when generated by AlphaFold 2, tFold or RoseTTAfold, but not I-TASSER.

Here, we provide the structure of a SctX-family protein. YscX and YscY depend on each other for soluble expression in *E. coli*, which is understandable given how entwined the two proteins are in their complex. Lacking YscX, YscY would present a large hydrophobic face at its N-terminal TPR that could explain why it is insoluble in isolation. After secretion of its substrate, this face might be responsible for recognizing SycD, an interaction previously observed[22,23]. We found that the N-terminal 31 residues of YscX are prone to proteolysis, indicating that they may not exhibit a tertiary structure. It is possible that the N-terminus carries a proteinaceous signal for YscX secretion, especially considering how conserved the sequence is among the SctX family (Fig. 2a). The smallest YscX fragment to direct secretion (amino acids 1–60) contains helix α1[17], leaving the possibility that YscY bound to helix α1 of YscX is part of the secretion signal. This does not seem to be the case for the structurally similar YscEFG complex, for which the transport of YscF to the injectisome depends on its co-chaperone YscE, which structurally fulfills the role of α2 and α3 in YscX[44]. Alternatively, the structurally unresolved N-terminus of YscX might bind other T3S proteins and only fold in their presence, as was observed for YopN in presence of TyeA in the same secretion system[45]. It is unclear whether YscX serves a hitherto unknown function outside the bacterial cell after its secretion or if its absence from the export apparatus signals a switch in substrate recognition, as was postulated by Day, et al.[17].

When compared to other T3S substrate:chaperone complexes, the structural similarities between YscXY and the complex of the needle protein with its two chaperones become obvious (Fig. 8a). Two structures of such a complex, one from *Pseudomonas* (PscEFG, PDB: 2UWJ[14]) and the other from *Yersinia* (YscEFG, PDB: 2P58[13]) have been published, both revealing an amphipathic SctF helix that binds into a hydrophobic groove formed by the concave side of the TPR protein YscG/PscG. The co-chaperone YscE/PscE was found to bind the hydrophobic face of the N-terminal TPR of YscG/PscG with its two helices that contribute several leucine residues to the binding interface. Thus, it appears as if YscX fulfills the roles of both "co-chaperone" and "substrate" in this assembly reminiscent of the YscEFG complex. In contrast, class II substrate:chaperone complexes—which are

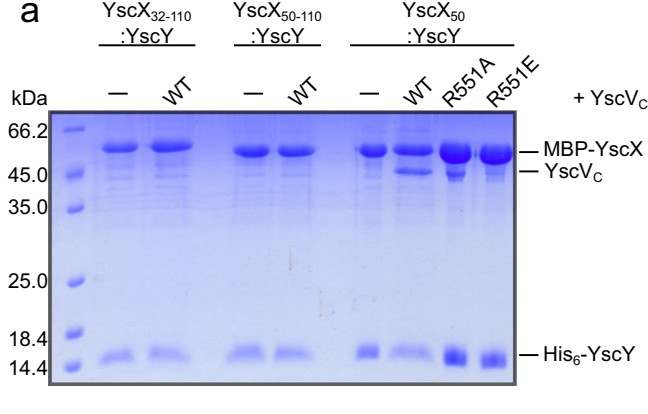

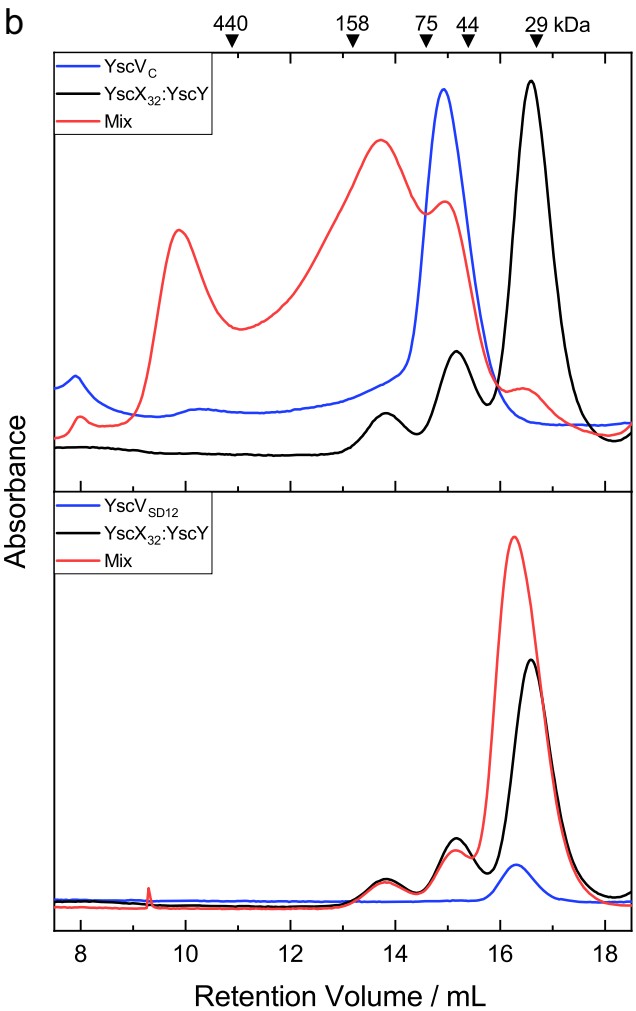

**Fig. 6 Recognition of YscXY by YscV$_C$ depends on the C-terminus of YscX as well as SD3 and SD4 of YscV$_C$. a** Coomassie-stained SDS-PAGE of MBP pull-downs. N- and C-terminally truncated MBP-YscX constructs were co-expressed with His$_6$-YscY and YscV$_C$. Lysates were loaded onto amylose resin and the washed matrix loaded onto the gel. The pull-downs generated consistent results over three replications. Source data are provided as a Source Data file. **b** Analytical size exclusion chromatography with 10 µM YscV$_C$, YscV$_{SD12}$ (both blue), YscX$_{32}$:YscY (black), or mixtures (red) of the proteins. Absorbance at 280 nm shows a shift in retention volume for a mixture of YscV$_C$ with YscX$_{32}$:YscY, but not YscV$_{SD12}$ with YscX$_{32}$:YscY. Chromatographies involving YscV$_{SD12}$ or YscV$_C$ were performed twice or four times, respectively, with consistent results. A SDS-PAGE and Western blot analyzing contents of gel filtration peaks are shown in Supplementary Fig. 8.

**Fig. 7 Concentration-dependent nonamerization of YscV$_C$.** For comparability, absorbances were normalized to the highest peak. From top to bottom: YscV$_C$, a ternary complex of YscV$_C$ co-expressed with YscX$_{50}$ and His$_6$-YscY, and the same ternary complex but with full-length YscX (YscX$_{fl}$) were loaded at 2.5 µM (pink) to 100 µM (blue) onto a Superdex 200 column. For reference, the expected retention volumes for the monomeric species (or 1:1:1 heterotrimer) and nonameric species (assuming 9:9:9 stoichiometry and a globular shape) are shown as dashed gray lines. The weak auto-nonamerization of YscV$_C$ is enhanced by the presence of YscX$_{50}$ but severely impeded by full-length YscX. Experiments were performed three times with consistent results.

hydrophobic translocator proteins and their chaperones—form soluble 1:1 heterodimers like YscXY, but exhibit vastly different binding behaviors. A short stretch close to the N-terminus of the substrate binds to the groove formed by the chaperone's concave side in extended conformation (Fig. 8b, refs. [8–11,46]). The structure of *Aeromonas* AcrH with translocator protein AopB shows further interactions between the two proteins. Analogous to the YscEFG and YscXY complex, AopB binds the N-terminal TPR of AcrH via an α-helix. In addition, AopB extensively wraps around the convex side of the chaperone, a feature neither present in YscXY nor YscEFG[11]. Thus, despite its heterodimeric nature, the

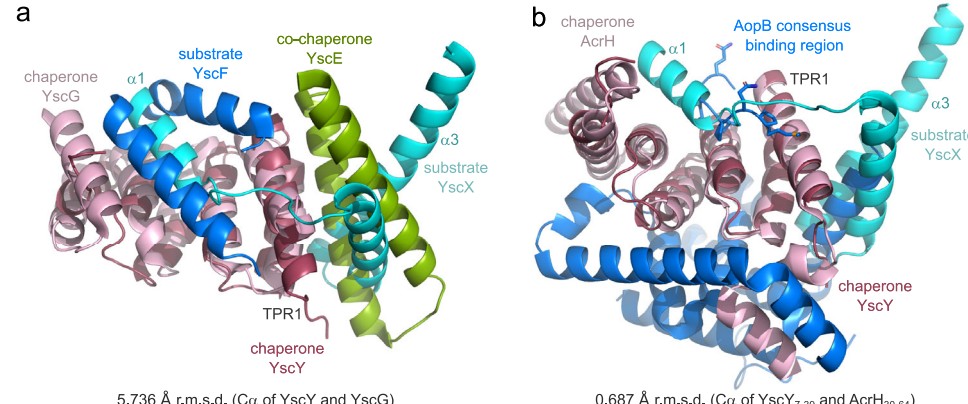

**Fig. 8 Comparison of YscXY with other T3S substrate:chaperone complexes. a** Superposition of YscX$_{50}$:YscY with YscEFG (PDB: 2P58). The alignment was done using the chaperones YscY (red) and YscG (pink) with an r.m.s.d. of 5.736 Å. The substrates YscX (cyan) and YscF (blue) as well as the co-chaperone YscE (green) are also shown. **b** Superposition of YscXY with the complex of class II chaperone AcrH (pink) and the major translocator protein AopB (blue), both from *Aeromonas* (PDB: 3WXX). The alignment was calculated using the first TPR of YscY (residues 7–39) and the first TPR of AcrH (residues 30–64) with an r.m.s.d. of 0.687 Å to obtain the right register. For clarity, sticks are shown for side chains of AopB where it binds the TPR groove of AcrH.

YscXY complex more closely resembles the heterotrimeric YscEFG complex, suggesting structural similarity to class III chaperones.

Since structures of FlhA were solved in 2010, several flagellar and injectisomal export gates were deposited in the PDB (FlhA:[26–28] SctV:[29–33,35]). Some discussion is dedicated to elucidating differences in how the subdomains are arranged relative to each other, especially considering the cleft between SD2 and SD4 that is responsible for the recognition of the flagellar FliD:FliT and FliC:FliS substrate:chaperone complexes[38]. The structure of YscV$_C$ was previously reported as adopting the "open" conformation with space between SD2 and SD4[35] when present without additional binding partners. Our structure of the ternary complex YscV$_C$:YscX$_{32}$:YscY exhibits the same open conformation showing that no major domain rearrangements are necessary to accommodate YscXY (Fig. 9a). Minor movements are observed at the binding site between YscY and SD4 as well as helix α3 of YscX and SD4 but these do not seem to affect the export gate globally (Supplementary Fig. 7). Our model also establishes that a highly conserved $^{46}$RLYP$^{49}$ motif of YscX binds within the aforementioned SD2-SD4 cleft. While we could model this region only in some of the chains, considerable difference electron density is present in the remaining chains indicating that this binding is a consistent feature. Interestingly, the binding site coincides with the recognition site of flagellar FliT and FliS chaperones (Fig. 9b, c). The binding of YscXY to YscV as seen in our structure might block access of other chaperone-bound secretion substrates to the SD2-SD4 cleft and thereby impede their secretion.

The major binding site for YscX lies between the fourth subdomain of two adjacent subunits of the YscV nonamer, where the C-terminal α3 helix of YscX inserts, pointing towards the inner cavity of the ring but not reaching it. While we initially thought that this hints at YscX stabilizing an otherwise dissociation-prone YscV$_C$ nonamer in vivo, our gel filtration results indicate that full-length YscX actually disfavors nonamerization of the export gate (Fig. 7 bottom). The importance of the YscX C-terminus for recognition by YscV explains why complementation of *yscX* deletion strains with C-terminally FLAG-tagged YscX does not produce an injectisome competent to secrete Yops[17]. The same construct, however, was secreted by a functional T3SS[17], suggesting that the interaction of YscXY with the SD2-SD4 cleft is sufficient for the secretion of YscX.

The binding site of the ATPase stalk protein SctO, as published for the complex of *Chlamydia* CdsV:CdsO[32], is likewise located in the SD4-SD4 cleft and would not allow for simultaneous binding of SctO and SctXY (Fig. 9d). It should be noted that the *Chlamydia* T3SS does not belong to the Ysc family and therefore lacks SctX and SctY. It is still poorly understood how different factors work together in the regulation of SctV. These include the binding of interacting proteins such as SctO, SepL:SepD in enteropathogenic *E. coli*[47], or substrate:chaperone complexes as well as local pH[33], and the linker between transmembrane and cytosolic domain[34,48,49]. As such, further studies should investigate the potential interplay between YscV, YscO, and YscXY. We argue that, based on the lack of rearrangement after YscXY binding and the nonamer destabilization in vitro, YscX (and its chaperone YscY) may contribute to those dynamics by destabilizing the nonameric ring of YscV, most likely via YscX's yet uncharacterized N-terminus. This may contribute to facilitating the substrate switch from needle components to hydrophobic translocators.

## Methods

**Cloning.** The vectors pETM-40_*yscX* and pACYCDuet-1_*yscY* were generated for the expression of YscX:YscY binary complexes via PCR and restriction cloning using *NcoI* and *NotI*. Furthermore, pACYCDuet-1 containing both *yscY* in its first and *yscV*356 in its second multiple cloning site was prepared using the same method but employing *NdeI* and *XhoI*. For the expression of YscV$_C$ alone, pETM-11_*yscV*356 was used. From this vector, a deletion mutant lacking SD3 and SD4 was generated (YscV$_{SD12}$). All genes were cloned from the *Yersinia enterocolitica* pYVe227 virulence plasmid (G. Cornelis, Basel) via PCR. Deletions and mutations were introduced via 'Round the horn mutagenesis. For a full overview of all constructs used, refer to Supplementary Table 3.

**Expression and purification of YscX:YscY complexes.** Complexes of YscX with its chaperone YscY were expressed in *E. coli* BL21 (DE3) transformed with both pETM-40_*yscX* and pACYCDuet-1_*yscY* vectors. Cell cultures were grown at 37 °C to OD$_{600}$ = 0.5 in LB medium and cooled to 20 °C before induction with 0.25 mM isopropyl β-D-1-thiogalactopyrsanoside at OD$_{600}$ = 0.8–0.9. After incubation over night, cells were spun down at 4000 × *g* and pellets stored at −20 °C. Cell pellets were resuspended in 50 mM Tris pH 8, 150 mM NaCl and 10 mM β-mercaptoethanol supplemented with one tablet of cOmplete protease inhibitor cocktail (Roche) and 0.6 mg DNase I per liter of expression culture and then passed through a pressure cell homogenizer (Stansted FPG12800) at 120 MPa.

Lysates were cleared by centrifugation (60 min, 30,000 × *g*, 4 °C). Amylose affinity chromatography (10 mL Amylose Resin HighFlow, New England Biolabs) was employed to capture MBP-YscX:His$_6$-YscY. After 60 min of light pivoting at 4 °C, the resin was washed by raising the NaCl concentration to 300 mM. The protein-loaded amylose was re-equilibrated with 50 mM Tris pH 8, 150 mM NaCl,

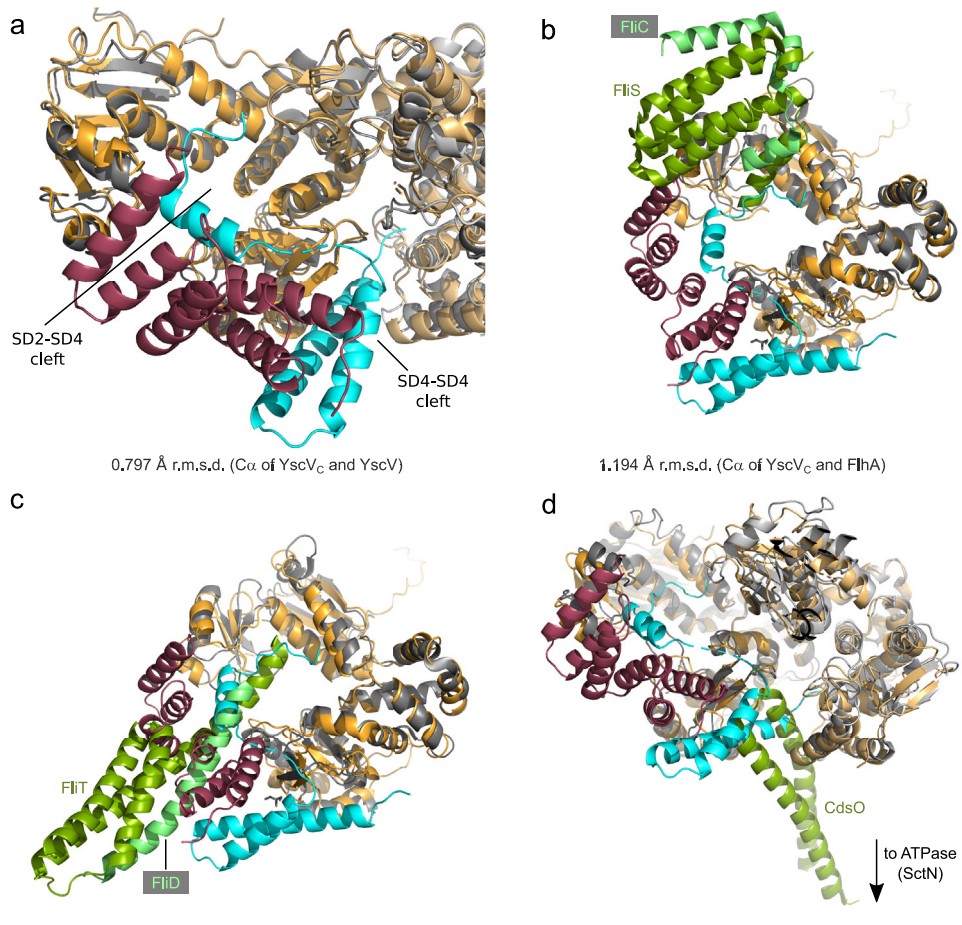

**Fig. 9 Superposition of export gate complexes with YscVXY.** YscV, YscX, and YscY are shown in orange, cyan, and red, respectively. **a** Superposition with the cryo-EM structure of YscV$_C$ (gray, PDB: 7ALW), which is shown as cartoon representation. YscV$_C$ protomers are aligned with an r.m.s.d. of 0.797 Å. **b**, **c** Superposition with FlhA (gray) bound to the flagellar chaperone:substrate complexes (green:light green). **b** FlhA:FliS:FliC (PDB: 6CH3). **c** FlhA:FliT:FliD (PDB: 6CH2). Alignments were obtained using SD3 of YscV$_C$ (residues 514–600) with an r.m.s.d. of 1.194 Å (PDB: 6CH2) and 1.207 Å (PDB: 6CH3). **d** YscVXY superimposed with *Chlamydia pneumoniae* CdsV in complex with the stalk protein CdsO (PDB: 6WA9). The alignment was calculated using SD3 of YscV$_C$ with an r.m.s.d. of 0.854 Å. Two subunits of YscV and CdsV (gray) are shown, CdsO is colored in green.

and 1 mM dithiothreitol. To remove the MBP-tag of YscX, 1 mg of TEV-protease was added per 50 mg of target protein and the resin was incubated at 20 °C over night on a seesaw shaker. The flow-through and successive washing fractions were collected, pooled and imidazole was added to a final concentration of 15 mM before being loaded onto 5 mL of NiNTA resin (Protino Ni-NTA Agarose, Macherey-Nagel).

After 60 min of incubation, unbound protein was washed from the column by increasing the imidazole and NaCl concentrations to 100 and 300 mM, respectively. YscX:YscY eluted from the resin when adding elution buffer (20 mM Tris pH 8, 150 mM NaCl, 250 mM imidazole, 1 mM DTT). Finally, the protein was purified via size exclusion chromatography on a HiLoad 16/60 Superdex 75pg (Cytiva) where the buffer was exchanged to 20 mM Tris pH 8, 150 mM NaCl. After concentrating the complex to approximately 10 mg/mL, 5 mM tris(2-carboxyethyl) phosphine (TCEP) was added and the solution was flash-frozen at −80 °C.

**Reductive methylation of YscX:YscY.** NiNTA-purified protein was first dialyzed against 2×2 L 50 mM Hepes pH 8, 300 mM NaCl, 2 mM DTT and then diluted to approximately 1 mg/mL. Borane dimethylamine complex (97%, Alfa Aesar) as well as formaldehyde were added to 1.2 mg/mL and 1.2% (v/v), respectively, and the mixture incubated on ice. After three and six hours, the same amounts of the two reagents were added and the protein was further incubated over night. The reaction was stopped with the addition of 250 mM Tris pH 8, the protein was concentrated and purified via gel filtration as described above.

**Expression and purification of YscV$_C$XY.** For the ternary complex of YscX$_{32}$:YscY with YscV$_C$, pETM-40_yscX32 and pACYCDuet-1_yscY_yscV356 were co-transformed into *E. coli* BL21 (DE3). Expression and purification steps were identical until the protein was eluted from the amylose resin after TEV-digestion. Instead of successive NiNTA affinity chromatography, the complex was loaded

onto 5 mL amylose resin to remove the remaining MBP-tagged protein. The flow-through and washing fractions were combined, concentrated, and purified via size exclusion chromatography on a HiLoad 16/60 Superdex 200 pg (Cytiva). Before freezing the concentrated protein complex, 5 mM TCEP was added.

**Expression and purification of His$_6$-YscV constructs.** For the stand-alone purification of the cytosolic domain of YscV (YscV$_C$, residues 356–704) or YscV$_{SD12}$ (residues 356–513), the proteins were expressed as described above from pETM-11 with an N-terminal and TEV-cleavable hexahistidine tag. The cleared lysate was loaded onto 10 mL NiNTA resin and incubated for 60 min with 10 mM imidazole. The flow-through was collected and the resin was washed first with 50 mM Tris pH 8, 300 mM NaCl, 10 mM imidazole, and 10 mM β-mercaptoethanol and then with 30 mM imidazole in the same buffer. An increase to 250 mM imidazole eluted the target protein, which was then dialyzed against 20 mM Tris pH 8, 150 mM NaCl, 1 mM DTT after adding 1:50 w/w TEV protease. Re-chromatographic NiNTA cleared the target of uncleaved His$_6$-YscV before the protein was concentrated and polished via size exclusion chromatography on a HiLoad 16/60 Superdex 200pg (YscV$_C$) or HiLoad 16/60 Superdex 75pg (YscV$_{SD12}$) (Cytiva) with 20 mM Tris pH 8, 150 mM NaCl as running buffer. After concentration to 10–15 mg/mL, 5 mM TCEP was added and the protein was flash-frozen.

**MBP-pull-down assay.** To test the binding of different YscX truncations to YscV$_C$, both proteins and YscY were co-expressed as described for the ternary complex but at a scale of 100 mL culture. The cleared lysate was applied to 1 mL of amylose resin and unbound protein was washed off. Detection of amylose-bound protein was achieved by resuspending the resin in washing buffer and loading it on an SDS-PAGE.

**Limited proteolysis**. YscX$_{full-length}$:YscY was subjected to limited proteolysis by trypsin, chymotrypsin, proteinase K, and thermolysin to identify stable constructs for expression. Purified protein (1 mg/mL) was mixed with 1:100 (*w/w*) protease in 50 mM Tris pH 8, 150 mM NaCl, and 1 mM DTT or 0.5 mM CaCl$_2$ (for thermolysin) and incubated on ice for 3 h. Samples were analyzed by SDS-PAGE and sequenced at their N-terminus via Edman degradation (Proteome Factory, Berlin, DE) after transfer to PVDF membrane (Immobilon, Merck) in a semi-dry set-up. Total mass was determined in-house using MALDI-TOF after dialysis against 10 mM acetate buffer pH 4.6.

**Analytical size exclusion**. Size exclusion chromatography on a Superdex 200 10/300GL was utilized to analyze protein oligomerization, dissociation, and association. The proteins were thawed on ice, combined for protein mixtures, or diluted for concentration gradients, and incubated for at least 30 min on ice before centrifugation and application to the column. All runs were carried out at 0.5 mL/min flowrate with 20 mM Tris pH 8, 150 mM NaCl as running buffer.

**Crystallization and data collection**. Crystals of YscX$_{50}$:YscY grew in 0.1 M BisTris pH 6.0, 0.2 M MgCl$_2$ and 21% PEG 3,350 with 5 mg/mL protein and a drop ratio of 0.33 µL reservoir to 0.66 µL protein solution at 20 °C in a sitting drop setting over 60 µL of reservoir. Microseeding of these into the same condition but with 13% PEG 3350 and 3–5 mg/mL protein was necessary to obtain single crystals. Drops were equilibrated for 6 h or over night before adding the seeds. Crystal growth was poorly reproducible and depended on purification batch. For cryoprotection, the crystals were transferred to a drop containing 0.1 M BisTris pH 6.0, 0.2 M CaCl$_2$, 19% PEG 3350 supplemented with 5% glycerol. Measurements were carried out at the BL14.2. beamline at the BESSY II electron storage ring operated by the Helmholtz-Zentrum Berlin für Materialien und Energie[50]. Data were collected using the local installation of MxCube[51].

For crystallization of the methylated complex YscX$_{32}$:YscY, the protein buffer was exchanged to 10 mM Tris pH 8, 50 mM NaCl, 5% (w/v) sucrose, 2.5 mM TCEP. Crystals were obtained at 20–25 °C with 0.1 M sodium succinate pH 6–6.5, 4.2–4.4 M NaCl as reservoir solution using 5 mg/mL protein but could be optimized by the addition of 4% acetonitrile (Hampton Research Additive Screen). The drop ratio was 1 µL protein plus 0.5 or 1 µL reservoir solution. For harvesting, trays were transferred from 20 to 4 °C, and crystals were cryoprotected with 10 % (w/v) sucrose. Data collection took place at beamline X06SA at SLS (Villigen, Switzerland) using 1.00 Å wavelength and 0.25° of oscillation between frames.

The ternary complex of YscV$_C$, YscX$_{32}$, and His$_6$-YscY was crystallized in a sitting drop setting with 0.1 M Hepes pH 6.8–7.5, 1% PEG 2000 MME, and 0.7–1.0 M sodium succinate as reservoir solution. The drop contained 0.33 µL reservoir solution and 0.66 µL of 5 mg/mL protein. Large crystals between 200 and 1000 µm in length grew after 2–3 days of incubation at 20 °C. As these crystals were very fragile and tended to crumble when touched, the plates were transferred to 4 °C the evening before harvesting. Cryoprotection was achieved by gradually transferring the crystals to solutions containing 0.1 M Hepes, 1% PEG 2000 MME, and 1.1 M sodium succinate as well as 5, 10, 15, and finally 22.5% glycerol. Data were collected through the local installation of MxCube[51] at beamline P13 at DESY (Hamburg, Germany) using a helical scan with 0.066 s exposure time per image and 100% transmission[52].

**Data reduction, model building, and refinement**. Diffraction data were indexed, integrated, and scaled using XDS[53] or XDSAPP[54] for YscX$_{50}$:YscY. The MR model was generated using a local installation of AlphaFold 2[43], supplying the sequences of YscX and YscY with a 33 residue linker between them. The phase problem was initially solved for the YscX$_{50}$:YscY data with a truncated model of the complex using Phaser[55] with the tNCS option disabled. After initial model building using BUCCANEER[56] in the CCP4i GUI[57] successive rounds of manual model building in Coot[58] and refinement in phenix.refine[59,60] generated the final model. TLS parameters were determined by the "find TLS" option in phenix.refine.

For YscX$_{32}$:YscY a model of the shorter YscX$_{50}$:YscY construct was employed as a search model during MR. Secondary structure restraints and reference model restraints from the aforementioned structure were used during refinement in phenix.refine. The ternary complex structure was solved via MR with a cryo-EM model of YscV$_C$[35] (PDB: 7ALW (YscV$_C$ nonamer)). After an initial round of refinement in phenix.refine, the YscX and YscY chains were placed manually into the density and refined with reference model restraints. To account for the low resolution data, group B-factors, secondary structure restraints, Ramachandran restraints, and torsion-angle NCS restraints were enabled in refinement. The resolution cutoff for the YscV$_C$X$_{32}$Y dataset was determined using paired refinements[61] as implemented in PAIREF[62,63].

Molecular graphics and alignments were generated in PyMOL. The r.m.s.d. value of alignments was calculated in PyMOL using only Cα atoms without outlier rejection. The entire residue range was considered unless stated otherwise.

**Reporting summary**. Further information on research design is available in the Nature Research Reporting Summary linked to this article.

## Data availability
The atomic coordinates and X-ray intensities generated in this study have been deposited in the Worldwide Protein Data Bank (wwPDB) under accession codes 7QIH (YscX$_{50}$:YscY), 7QII (YscX$_{32}$:YscY), and 7QIJ (YscV$_C$:YscX$_{32}$:YscY). Diffraction images are available from the SB Grid Data Bank[64] (data.sbgrid.org) with Data ID 905 (YscX$_{50}$:YscY), Data ID 906 (YscX$_{32}$:YscY), and Data ID 907 (YscV$_C$:YscX$_{32}$:YscY). The structure of the YscV$_C$ nonamer used as search model to solve the phase problem of the YscVXY structure is available from the wwPDB under accession code 7ALW. The structures used to generate Figs. 8 and 9 are available from the wwPDB. Figure 8a: 2P58 (YscEFG); Fig. 8b: 3WXX (AcrH:AopB); Fig.9a: 7ALW (YscV$_C$ nonamer); Fig. 9b: 6CH3 (FlhA:FliS:FliC); Fig. 9c: 6CH2 (FlhA:FliT:FliD); Fig. 9d: 6WA9 (CdsV:CdsO). Source data are provided with this paper.

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

## Acknowledgements

We thank the Helmholtz-Zentrum für Materialien und Energie for the allocation of synchrotron radiation beamtime. We acknowledge the Paul Scherrer Institut, Villigen, Switzerland for provision of synchrotron radiation beamtime at beamline X06SA (PX1) of the SLS and would like to thank Tomizaki Takashi for assistance. The synchrotron data were collected at beamline P13 operated by EMBL Hamburg at the PETRA III storage ring (DESY, Hamburg, Germany). We would like to thank Isabel Bento for the assistance in using the beamline. We acknowledge the help of Dr. Jens Sproß (Bielefeld University) in preparing, collecting, and analyzing mass spectrometry data as well as Dr. Gregor Hagelueken (University of Bonn) for generating the AlphaFold 2 models. We thank Martin Malý (Czech Technical University) for his help in running PAIREF. D.G. acknowledges funding from the Bielefelder Nachwuchsfonds. H.H.N. thankfully acknowledges the financial support by HZB (travel grant "BESSY 2 MX: 17").

## Author contributions

H.H.N., D.G. and M.S. conceived the project. D.G. and M.S. performed experiments. D.G., M.S. and H.H.N. collected and analyzed data. D.G. prepared the figures. D.G. and H.H.N. wrote the manuscript. H.H.N. supervised the project.

## Funding

## Competing interests

The authors declare no competing interests.
