## [Peer Review File · Nature Communications]

Direct interaction of a chaperone-bound type III secretion substrate with the export gateREVIEWER COMMENTS

Reviewer #1 (Remarks to the Author):

Gilzer et al. present a rigorous structural analysis of the YscXY complex and YscXYV complex. There are no obvious mechanistic insights and the authors do not make any claims that aren't supported by their structures. The manuscript is essentially descriptive in nature. This is not meant as a criticism.

The main result is the recognition that YscY is similar to YscG. They do not appear to share sequence identity, so this is a significant finding. The other important finding is that YscXY binds the SD 2-4 cleft. This is similar, but not identical, to the results from Kalodimos, who showed that FlhS and FlhT bind in a similar location. These are very different systems, so it isn't really worthwhile to focus on the differences, and they don't. The binding in the SD4-SD4 cleft is a bit more interesting, to me, this interaction is incompatible with the binding of CdsO, as they note. This is intriguing, and I take it to mean that the 4/4 cleft on the other side of XY could be occupied by the lever arm and receive force from the ATPase. I can understand why the authors don't speculate on this in their discussion.

The main contributions are the YscXY structures and seeing cargo, in a different system, bind to the SD2/4 cleft.

There are some minor improvements that can be made for clarity, but they do not alter the findings or significance.

The largest concern is that I fear they have overstated the resolution of the VXY complex. The validation report shows that this structure compares poorly to other 3.7Å structures. Table 1 indicates a $cc1/2$ of 0.394 in the highest bin. This is well within the acceptable range, but these data do not seem to produce a model with expected statistics. I would urge the authors to verify that the high-resolution data are helping the model. The L test in the validation report indicates that twinning is a concern. Phenix can find this automatically, but it isn't clear from the methods how or if this was dealt with during refinement.

The density figures in S3 really aren't clear and do not include the YscV ring. Also, it isn't clear that they are correctly described. They are described as unweighted maps, which would be very noisy with these data. Are these 2mFo-DFc maps?

It would be interesting to see the predicted models for the YscXY complex. They note that Alpha fold correctly positioned the n-term helix, but don't show this in S1.

Please give RMSD's and residue ranges throughout when presenting aligned structures.

The first two paragraphs of the results seem to belong in the methods. The importance of models in the structure determination is interesting, but it doesn't seem to be a result.

Figure 4 should have some or all residues mentioned in the text labeled in the figures (B and C). The legend should use the site A, B, C terminology presented in the text.

Figure 6 should include gels showing the contents of the peaks. Panel A. should have the ladder molecular weights labeled.

Figure 7 and the section headed Nonamerization of YscV may be regulated by YscX, really doesn't add anything to the manuscript. YscV is missing the entire membrane-spanning region.

Good luck with your submission,

Ben Spiller

Reviewer #2 (Remarks to the Author):

The T3SS is a complex, macromolecular protein export machinery, which is part of two evolutionary related bacterial nanomachines: the injectisome and the bacterial flagellum. The injectisome is a needle-like structure which enables the bacteria to directly inject effector proteins into host cells. The export gate, an important element of the T3SS, mediates substrate recognition and export among other functions.

In the present manuscript by Gilzer et.al., the authors present structures of a substrate-chaperone-complex (YscX-YscY) of *Yersinia* and the substrate-chaperone-complex bound to the export gate, specifically a nonameric ring formed by YscVC. They identify binding sites between YscX and YscY, as well as between the complex and YscVC and compare those binding sites to other injectisomal and flagellar substrate:chaperone complexes. Interestingly, even though the heterodimeric nature is comparable to injectisomal class II chaperones, the binding behavior is very similar to class III chaperones. This indicates that the substrate YscX might fulfill a second role as "co-chaperone" even though the exact role of this protein remains obscure.

In conclusion, the current manuscript provides the readers with a first structure of a YscXY complex, a specific substrate-chaperone pair of the Ysc/Yop-T3SS-family, docked at the export gate. The structural data provided here will help further research to investigate the function of the YscXY complex in the assembly of the injectisome.

Comments:

- 1) Fig. 1b+c and Fig. 5: Why show a surface representation of the electrostatic potential and not a hydrophobicity gradient as discussed in the text? I suppose that red indicates negative and blue positive potential, but this should be mentioned in the figure or legend.
- 2) Fig. 4a: For clarity, YscY should be labelled in the figure.
- 3) Fig. 4e: The SD4-SD4 cleft is never mentioned in the text and should be clarified, because the cleft is also mentioned again in Figure 9a. It is also a little confusing to state that the "involved" YscVC subunits shown are SD3 and SD4.
- 4) Fig 6a: Is the SDS-PAGE Coomassie stained? It is a little bit confusing, that the authors are using the term "YscX50-122" in the figure and "YscX50" in the text but mean the same thing. Should be clarified or rephrased because it complicates understanding of the data. The molecular masses of the protein standard should be shown.
- 5) Fig. 7, lines 378ff, lines 467ff: How can the conclusion from the size exclusion experiment (Fig. 7), i.e. that nonamerization of YscV is regulated by YscX, be reconciled with the fact that YscV homologs can assemble into a nonameric structure on its own in other species and also YscV assembles in vivo in YscX/YscY mutant strains in *Yersinia* (10.1111/j.1365-2958.2012.08146.x)?
- 6) Fig. 8: Consider re-coloring YscG and YscY. Pink and red are hard to distinguish. The figures would be easier to read if all proteins would be labelled.
- 7) The nomenclature should be harmonized throughout the manuscript e.g. the author switch in the middle of the introduction from using the Sct nomenclature to the *Yersinia* protein names of YscX and YscY.
- 8) Line 24: vT3SS – what does the v stands for? I assume virulence associated, but it should be clarified in the text.
- 9) Line 24/25: T3SSs are not a mechanism, they are the machineries executing a mechanism. I would advise to rephrase the first sentence.
- 10) Line 28: fT3SS abbreviation not used correctly. A flagellar type III secretion system is not the same as but rather a part of the bacterial flagellum.

- 11) Line 47/48: "even though significant binding occurs also on the convex side of the chaperone" – might be a clearer phrasing
- 12) Line 48: typo; correct to "gatekeeper"
- 13) Line 86: virulence associated T3SS might be a better phrasing than "pathogenic T3SS"
- 14) Line 243: why not use the abbreviation T3SC?
- 15) Line 260: typo; there is one "the last" too many
- 16) Line 306-307: The authors should decide between the three or one letter amino acid code
- 17) Line 353: Clarify which amino acids are residues 47 and 48
- 18) Line 357: C-terminal instead of last would be a better phrasing
- 19) Line 379: typo: "C-terminus" instead of "C-Terminus"
- 20) Line 469: typo: "yxcX deletion"
- 21) Fig. 6 legend, Line 699: Typo: change "bei" to "by"

Reviewer #3 (Remarks to the Author):

The manuscript by Gilzer et al. provides the first structural information for the SctX-SctY (YscX-YscY) family of proteins that are unique, but essential components, to all of the members of the Ysc family of T3SSs. YscY is a TPR-type chaperone that directly interacts with the secreted YscX protein. The investigators also present initial structural information on the interaction of the YscX-YscY complex with the SctV (YscV) component of the export apparatus. These structures suggest that the YscX-YscY complex interacts with YscV in two ways: (1) a conserved interaction between the N-terminal region of YscX and YscY with an SD2-SD4 cleft that is similar in location to the region of SctV (FlhA) involved in the interaction with flagellar substrate-chaperone complexes; and (2) what appears to be a unique interaction between the essential YscX C-terminal region and the SD4 domains of adjacent YscV protomers. The conserved interaction likely represents an interaction involved in targeting YscX for secretion; whereas, the unique interaction likely points to the unique function of YscX (and YscY) in the T3S process. Functional studies (pull downs and chromatography) confirm the YscX-YscY interaction, and role of the YscX C-terminal region, with YscV. Furthermore, chromatography studies suggest the N-terminal region of YscX plays an important role in regulating the multimerization of the YscV nanomer. Overall, the structural and functional results presented hash well with previous results presented on the function of YscX, YscY and YscV in the T3S process and this information is well discussed in relation to the structural information provided. Overall, the manuscript presents novel and noteworthy results that will be of significant interest to the T3SS and flagellar fields of study.

The results presented support the conclusions presented by the investigators. Importantly, the results and methods provide information that will facilitate further investigations into the mechanism by which YscX-YscY and YscV contribute to the T3S process.

Additional comments:

1. Structural comparisons (Fig. 8) suggest that YscY is a class 3 chaperone; however, this is not stated.
2. Line 260-261. the last the last...
3. Fig. 6b. Line 368-371. The authors suggest the large peak seen in the ternary complex might represent a 1:1:1 ternary complex, which is possible, but was a 1:1:2 complex considered as YscX-YscY interacts with 2 adjacent YscV monomers?

Reviewer #1 (Remarks to the Author):

Gilzer et al. present a rigorous structural analysis of the YscXY complex and YscXYV complex. There are no obvious mechanistic insights and the authors do not make any claims that aren't supported by their structures. The manuscript is essentially descriptive in nature. This is not meant as a criticism.

The main result is the recognition that YscY is similar to YscG. They do not appear to share sequence identity, so this is a significant finding. The other important finding is that YscXY binds the SD 2-4 cleft. This is similar, but not identical, to the results from Kalodimos, who showed that FliS and FliT bind in a similar location. These are very different systems, so it isn't really worthwhile to focus on the differences, and they don't. The binding in the SD4-SD4 cleft is a bit more interesting, to me, this interaction is incompatible with the binding of CdsO, as they note. This is intriguing, and I take it to mean that the 4/4 cleft on the other side of XY could be occupied by the lever arm and receive force from the ATPase. I can understand why the authors don't speculate on this in their discussion.

The main contributions are the YscXY structures and seeing cargo, in a different system, bind to the SD2/4 cleft.

There are some minor improvements that can be made for clarity, but they do not alter the findings or significance.

We thank Dr. Spiller for his time and for the positive assessment of our work.

The largest concern is that I fear they have overstated the resolution of the VXY complex. The validation report shows that this structure compares poorly to other 3.7Å structures. Table 1 indicates a $cc1/2$ of 0.394 in the highest bin. This well within the acceptable range, but these data do not seem to produce a model with expected statistics. I would urge the authors to verify that the high-resolution data are helping the model.

We thank Dr. Spiller for this thoughtful comment. As he notes, we were guided by the \$CC1/2\$ value when choosing the high resolution cutoff. Following Dr. Spiller's comment, we now used paired refinements as originally described by Karplus and Diederichs (Science, 2012, doi:10.1126/science.1218231.), which allow determining whether inclusion of weak, high resolution data in refinements does actually improve the model or not. To this end, we used the software tool PAIREF, which had to be modified to work for our structure. We are very grateful to Martin Malý, the author of PAIREF, for adapting PAIREF and also for carrying out the calculations. Starting from a model refined against 4.2 Å data, he included higher resolution data in 0.1 Å or 0.2 Å steps and checked changes in \$R_{work}\$ and \$R_{free}\$ for data to only the previous resolution. The 0.2 Å step from 4.2 Å to 4.0 Å improved the model, while inclusion of higher resolution data did not (Fig. 1 and Table 1).

Fig. 1 Changes in R_{work} and R_{free} reported by PAIREF upon increasing resolution in steps of 0.2 Å (based on values given in Table 1). For each incremental step of resolution from X→Y, the pair of bars gives the changes in overall R_{work} and R_{free} for the model refined at resolution Y with respect to those for the model refined at resolution X, with both R values calculated at resolution X.

Table 1 Changes in R_{work} and R_{free} reported by PAIREF upon increasing resolution in steps of 0.2 Å. Improvements (i.e. negative differences between starting and final) are highlighted in green.

# Shell	R_{work}			R_{free}		
	init	fin	diff	init	fin	diff
4.20A->4.00A	0.3037	0.3026	-0.0011	0.3196	0.3187	-0.0009
4.00A->3.80A	0.3089	0.3118	0.0029	0.3245	0.3254	0.0009
3.80A->3.70A	0.3186	0.3183	-0.0003	0.3319	0.3325	0.0006

Analysis in 0.1 Å steps showed that the step from 4.2 to 4.1 Å improved the model, while inclusion of higher resolution data did not (Fig 2 and Table 2).

Fig. 2 Changes in R_{work} and R_{free} reported by PAIREF upon increasing resolution in steps of 0.1 Å (based on values given in Table 2).

Table 2 Changes in R_{work} and R_{free} reported by PAIREF upon increasing resolution in steps of 0.1 Å. Improvements (i.e. negative differences between starting and final) are highlighted in green.

# Shell	R_{work}			R_{free}		
	init	fin	diff	init	fin	diff
4.20A->4.10A	0.3037	0.3022	-0.0015	0.3196	0.3188	-0.0008
4.10A->4.00A	0.3055	0.3082	0.0027	0.3226	0.3238	0.0012
4.00A->3.90A	0.3112	0.3112	0.0000	0.3260	0.3266	0.0006
3.90A->3.80A	0.3146	0.3148	0.0002	0.3294	0.3297	0.0003
3.80A->3.70A	0.3181	0.3146	-0.0035	0.3329	0.3329	0.0000

Based on paired refinements, 4.1 Å appears to be the best high resolution cutoff. The model remained essentially unchanged for all tested high resolution cutoffs. We now deposited a model refined against 4.1 Å data in the PDB and changed Supplementary Table 1 and the text accordingly. We provide a new PDB validation report. Upon re-submitting data, PDB processing can take some time. Recently, we had to wait 17 days in one case. Therefore, we submit a preliminary PDB validation report for entry 7qij. We will provide the final report, as soon as we get it from the PDB.

The L test in the validation report indicates that twinning is a concern. Phenix can find this automatically, but it isn't clear from the methods how or if this was dealt with during refinement.

We are very grateful to Dr. Spiller for spotting this issue. The structure cannot simply be refined with a twin law, as no twinning is possible in this space group. If the structure really was twinned, it would have to be lower symmetry (e.g. $P2_1$) appearing as $P2_12_12_1$.

During data processing and structure determination, we never saw any indication of twinning, although we had used phenix.xtriage to analyze our data. The PDB also mentions Xtriage as source of

the values for $\langle |L| \rangle$ and $\langle L^2 \rangle$. Therefore, we re-ran phenix.xtriage on the file containing experimental data that we had uploaded to PDB in mtz format and on the structure factor file in cif format that the PDB generated during processing. With both files, we obtained values of $\langle |L| \rangle = 0.46$ and $\langle L^2 \rangle = 0.29$ which are close to the values expected for an untwinned structure (see Table 3). We contacted the PDB three times asking for assistance in resolving this issue, but the PDB never replied. We then ran several tests with phenix.xtriage on our own computer and the PDB validation server. We concluded that apparently the PDB validation performs the L-test for twinning starting from structure factor amplitudes instead of the intensities if both are present in the structure factor file (see Table 3). As the L-test is based on intensity statistics, this will produce less reliable results. By default, xtriage will use intensities and not amplitudes if a file contains both. When explicitly run with amplitudes as input, xtriage gives the following warning:

******WARNING***** Please be aware that the input data were given as amplitudes and squared for the purposes of this analysis, therefore the numbers displayed here are less reliable than the values calculated from the raw intensities.”*

We asked the PDB to look into this issue, but they replied that they are “not sure how long this process may take”. To obtain a more reliable result for the L-test in the PDB validation report, we therefore uploaded the experimental data to the PDB as intensities only. In this case, the PDB validation report produces the same values for $\langle |L| \rangle$ and $\langle L^2 \rangle$ (0.47 and 0.30, respectively) that we obtain when we run phenix.xtriage on our 4.1 Å mtz file containing both amplitudes and intensities (see Table 3).

In summary, we conclude that when performed properly on intensities, the L-test and all other twin indicators reported by phenix.xtriage (see Table 3) are very close to those expected for an untwinned structure. Therefore, we see no need to attempt twin refinements in a lower-symmetry space group.

Table 3: Values for $\langle |L| \rangle$ and $\langle L^2 \rangle$ from the PDB validation report (PDB) and when calculated with phenix.xtriage on our own computer. Expected values for untwinned crystals and a perfect twin are shown as reference.

	expected		3.7 Å data (initial submission)			4.1 Å data (revision)				
	Un- twinned	perfect twin	PDB	own	own	PDB	PDB	own	PDB	own
Xtriage run on (F or I)			default	F	I	default	F	F	I	I
Multivariate Z L-test	< 3.5			18.23	2.56			2.6		1.53
$\langle I^2 \rangle / \langle I \rangle^2$	2	1.5		2.00	2.10			2.07		2.07
$\langle F \rangle^2 / \langle F^2 \rangle$	0.785	0.885		0.83	0.79			0.80		0.79
$\langle E^2 - 1 \rangle$	0.736	0.541		0.67	0.73			0.73		0.73
$\langle L \rangle$	0.500	0.375	0.36	0.36	0.46	0.44	0.44	0.46	0.47	0.47
$\langle L^2 \rangle$	0.333	0.200	0.20	0.20	0.29	0.27	0.27	0.28	0.30	0.30

We want to point out that providing the data only as intensities resulted in a new processing error in the PDB validation report. Now, the validation report shows a data completeness of 91%. This value is wrong. However, we have not been able to track the source of this problem during PDB processing. Scaling with XSCALE showed a completeness of 99.6 %. We converted the data from XSCALE to mtz format with AIMLESS and then removed the amplitudes. The resulting mtz file containing only I, SIGI

and RFREE columns was uploaded to PDB, which converted it to CIF format. We ran phenix.xtriage over the mtz file uploaded to PDB and over the cif file generated by PDB. For both files, phenix.xtriage shows the following table for completeness.

-----Completeness (log-binning)-----
 The table below presents an alternative overview of data completeness, using the entire resolution range but on a logarithmic scale. This is more sensitive to missing low-resolution data (and is complementary to the separate table showing low-resolution completeness only).

Resolution	Reflections	Completeness
49.8347 - 38.5156	100/111	90.1%
38.3790 - 33.4021	100/102	98.0%
33.3292 - 28.9542	142/147	96.6%
28.9313 - 25.1607	227/233	97.4%
25.1313 - 21.8306	333/344	96.8%
21.8274 - 18.9651	504/509	99.0%
18.9549 - 16.4633	773/780	99.1%
16.4597 - 14.2962	1148/1157	99.2%
14.2949 - 12.4160	1751/1767	99.1%
12.4140 - 10.7815	2646/2671	99.1%
10.7810 - 9.3629	4018/4038	99.5%
9.3626 - 8.1309	6064/6090	99.6%
8.1306 - 7.0609	9229/9275	99.5%
7.0607 - 6.1320	14016/14045	99.8%
6.1318 - 5.3251	21289/21313	99.9%
5.3250 - 4.6244	32395/32450	99.8%
4.6244 - 4.1000	40643/40723	99.8%

Therefore, the data completeness given in the PDB validation report is obviously wrong. If the reviewers want to check themselves, we are happy to provide the file with the deposited intensities.

The density figures in S3 really aren't clear and do not include the YscV ring. Also, it isn't clear that they are correctly described. They are described as unweighted maps, which would be very noisy with these data. Are these 2mFo-DFc maps?

We thank Dr. Spiller for pointing this out. The maps are $2mF_o-DF_c$ maps, as is now stated in the legends to Fig. S3 and also Fig.5b.

For Fig. S3 we generated new figures with PyMOL to make the density clearer. We now also include density for the four subdomains of YscVc separately as new panels a-d.

It would be interesting to see the predicted models for the YscXY complex. They note that Alpha fold correctly positioned the n-term helix, but don't show this in S1.

Actually, we had included an overlay of our YscXY complex with the AlphaFold model in Fig. S1c of our initial submission. The figure shows that the N-terminal helix of YscX is correctly positioned when predicting the complex structure with AlphaFold2.

Please give RMSD's and residue ranges throughout when presenting aligned structures.

If not otherwise stated, the entire residue range of the aligned proteins was used without outlier rejection. We now explicitly mention this in methods (line 497-499 in version with changes tracked). We added RMSD values where they had been missing, e.g. in Figs. 1b+c, Fig. 8a+b, Fig. 9a-d, Fig. S1c and the legend to Fig. S7.

The first two paragraphs of the results seem to belong in the methods. The importance of models in the structure determination is interesting, but it doesn't seem to be a result.

The first two paragraphs of results mention the model used for structure determination in only one sentence and we assume that Dr. Spiller may be referring to the first paragraph of the discussion.

We definitely would like to keep the first two paragraphs of results. The first paragraph has nothing to do at all with structural models, but describes the rationale for the choice of construct boundaries. It briefly summarizes a substantial experimental effort. The second paragraph briefly describes aspects of crystallization and structure determination that we consider important. We think it is reasonable not to move this into methods. If Dr. Spiller or the editor request moving this section, we will be happy to comply.

We agree with Dr. Spiller that the paragraph dealing with the importance of models in structure determination is not strictly a result. Therefore, we had placed it in discussion right from the start. We think that it fits discussion better than methods and we would like to keep it in the discussion. Accurate structure prediction is a recent progress. In the last few months, several papers were published that only deal with the utility of models predicted by various servers for molecular replacement (e.g. Kryshtafovych et al. *Proteins*, doi 10.1002/prot.26223; McCoy et al. *Acta Crystallogr. D.* 2022, doi 10.1107/S2059798321012122; Millán et al. *Proteins* 2021, doi 10.1002/prot.26214) or with the impact of AlphaFold2 on protein crystallography and structural biology in general (e.g. Cramer *Nat. Struct. Mol. Biol.* 2021, doi 10.1038/s41594-021-00650-1; Jones and Thornton *Nat. Methods* 2022, doi 10.1038/s41592-021-01365-3; Masrati et al. *J. Mol. Biol.*, doi 10.1016/j.jmb.2021.167127) showing that this is a topic of current interest. Therefore, we think that it is justified to keep one paragraph discussing the impact of models on this particular project. We consider two aspects of our discussion particularly striking. First, YscY has been correctly predicted to have a TPR fold for a long time. Nevertheless, only the models predicted by the latest generation of structure prediction algorithms were sufficient to solve the structure. Second, we consider the good prediction of a protein complex with AlphaFold2 an interesting finding, particularly because only AlphaFold2 predicted the complex correctly.

Figure 4 should have some or all residues mentioned in the text labeled in the figures (B and C). The legend should use the site A, B, C terminology presented in the text.

We do not feel confident to show side chains for YscY due to their weak density. Therefore, we opted to highlight the corresponding C α -atoms. To this end, we changed from a cartoon to ribbon representation for YscX and YscY in panels b and c. For YscV, we now show side chains for presumably involved residues. We now also mention sites A-C in the figure legend to make cross-referencing with the text easier.

Figure 6 should include gels showing the contents of the peaks. Panel A. should have the ladder molecular weights labeled.

A replication of the runs with 10 μ M YscV, YscXY, or a mixture of both are shown in the new supplementary fig. 8. Due to the very low concentration of protein in fractions collected from these runs, we opted to show both a silver-stained gel – in which only YscV can be detected – and an anti-His Western Blot that detects His₆-YscY. Neither YscX nor YscY were visible in the silver-stained gel, which we argue is due to their small size and low concentration. Since YscY is visible in the Western blot, we infer that YscX must also be present in these fractions since neither of them can be maintained in solution alone.

Higher loading concentrations of the proteins could be used, but would result in a higher apparent molecular weight of YscV due to its dynamic assembly/disassembly and a less obvious shift upon binding YscXY. We did not include a gel showing contents of the SD12 or SD12 + YscXY runs because they (a) cannot be detected using silver staining and (b) only co-elute from the gel filtration due to their size and not interaction, as no shift in apparent weight was observed.

We added the molecular weight marker to Fig. 6.

Figure 7 and the section headed Nonamerization of YscV may be regulated by YscX, really doesn't add anything to the manuscript. YscV is missing the entire membrane-spanning region.

We concede that our constructs studied by gel filtration lack the membrane-spanning region. However, we do not think that this renders the results irrelevant. SctVs are large and complex proteins. Studying their oligomerization and the factors that influence it is anything but easy. Using reductionist approaches to address certain aspects of complex systems is standard in science. Studying individual domains of multidomain proteins is a common approach in biochemistry and molecular biology.

Of course, membrane-bound full-length YscV will behave differently than its isolated cytoplasmic domain. Nevertheless, our gel filtration data reveal several interesting aspects. They show that the binding of the YscX C-terminus to the SD4-SD4 cleft substantially stabilizes the YscV_c nonamer. They also show that the unstructured N-terminus of YscX completely offsets this effect resulting in disruption of the nonamer. To better reflect these points, we changed the heading of this section to “The termini of YscX oppositely influence nonamerization of YscV_c”. We also changed the introductory sentences to this paragraph to provide a clearer rationale for our gel filtration experiments (lines 260-268 in version with changes tracked).

We do not imply that YscX can also disassemble the nonamer of full-length YscV in bacteria. Nevertheless, our in vitro data indicate that even in bacteria, the binding of YscX might destabilize the nonameric ring of the YscV cytoplasmic domain. This destabilization upon YscX binding or stabilization upon YscX secretion might be important for the binding of other substrates. This conclusion could most likely not be obtained by using fluorescently tagged full-length YscV and brightness analysis of intact bacteria. Our gel filtration data may inspire further experiments that will hopefully allow deducing how exactly YscX exerts its essential function within Ysc family T3SS.

Good luck with your submission,

Ben Spiller

Reviewer #2 (Remarks to the Author):

The T3SS is a complex, macromolecular protein export machinery, which is part of two evolutionary related bacterial nanomachines: the injectisome and the bacterial flagellum. The injectisome is a needle-like structure which enables the bacteria to directly inject effector proteins into host cells. The export gate, an important element of the T3SS, mediates substrate recognition and export among other functions.

In the present manuscript by Gilzer et.al., the authors present structures of a substrate-chaperone-complex (YscX-YscY) of *Yersinia* and the substrate-chaperone-complex bound to the export gate, specifically a nonameric ring formed by YscVC. They identify binding sites between YscX and YscY, as well as between the complex and YscVC and compare those binding sites to other injectisomal and flagellar substrate:chaperone complexes. Interestingly, even though the heterodimeric nature is comparable to injectisomal class II chaperones, the binding behavior is very similar to class III chaperones. This indicates that the substrate YscX might fulfill a second role as “co-chaperone” even though the exact role of this protein remains obscure.

In conclusion, the current manuscript provides the readers with a first structure of a YscXY complex, a specific substrate-chaperone pair of the Ysc/Yop-T3SS-family, docked at the export gate. The structural data provided here will help further research to investigate the function of the YscXY complex in the assembly of the injectisome.

We thank Reviewer #2 for his time, the careful proofreading and for the positive assessment of our work.

Comments:

1) *Fig. 1b+c and Fig. 5: Why show a surface representation of the electrostatic potential and not a hydrophobicity gradient as discussed in the text? I suppose that red indicates negative and blue positive potential, but this should be mentioned in the figure or legend.*

We thank Reviewer #2 for this good suggestion. We switched in Fig. 1b+c and Fig. 5 to a hydrophobicity map of YscY and YscV, respectively. This makes the result even clearer than the electrostatic potential did. We changed the figure legends accordingly and cited Eisenberg *et al* (1984) for developing the used scale.

2) *Fig. 4a: For clarity, YscY should be labelled in the figure.*

We agree. We now labelled YscY.

3) *Fig. 4e: The SD4-SD4 cleft is never mentioned in the text and should be clarified, because the cleft is also mentioned again in Figure 9a. It is also a little confusing to state that the “involved” YscVC subunits shown are SD3 and SD4.*

We thank Reviewer #2 for pointing this out.

We now explain the SD4-SD4 cleft in the result section (lines 218-219 in version with changes tracked) and use the same term in the discussion section.

In the legend to Fig. 4e, the phrase “Both involved YscVc subunits” refers to two neighboring YscVc chains in the nonameric ring and not to subdomains SD3 and SD4 within one protomer. SD3 is also colored in Fig. 4e due to its involvement in charge neutralization between YscX’s C-terminus and YscV’s R551 (which is in SD3). To point this out, we added “SD3” in a sentence in results (line 220 in version with changes tracked) that now reads: “This is the major binding interface between YscV and YscX and involves SD3 and SD4 of YscV.”

4) Fig 6a: Is the SDS-PAGE Coomassie stained? It is a little bit confusing, that the authors are using the term “YscX50-122” in the figure and “YscX50” in the text but mean the same thing. Should be clarified or rephrased because it complicates understanding of the data. The molecular masses of the protein standard should be shown.

Yes, the SDS-PAGE is Coomassie stained. We added the staining in the figure legend.

We now changed the label in panel a from YscX50-122 to YscX50, as used in the text.

We now show molecular masses of the standard proteins.

5) Fig. 7, lines 378ff, lines 467ff: How can the conclusion from the size exclusion experiment (Fig. 7), i.e. that nonamerization of YscV is regulated by YscX, be reconciled with the fact that YscV homologs can assemble into a nonameric structure on its own in other species and also YscV assembles in vivo in YscX/YscY mutant strains in Yersinia (10.1111/j.1365-2958.2012.08146.x)?

We thank Reviewer #2 for pointing out that our statement regarding this experiment had not been sufficiently clear. The last comment of Reviewer #1 goes into a similar direction. Therefore, our reply is partly repetitive.

First, our experiments only contain the cytosolic domain of YscV. With regard to this, our heading was not sufficiently precise. Therefore, we changed “YscV” to “YscVc” in the heading.

Second, by “regulation of nonamerization”, we did not mean to say that YscX is essential for YscV nonamerization – neither in vitro, nor in bacteria. As Reviewer #2 points out, YscV can assemble in *Yersinia* lacking YscX and YscY. We now explicitly mention this in the introductory sentence to this paragraph of results (lines 263-264 in version with changes tracked). Our Fig. 7 shows that at sufficiently high concentration, the cytosolic domain of YscV forms nonamers in the absence of YscXY. Therefore, we see no contradiction between our gel filtration data and the in vivo data obtained by fluorescence microscopy.

To us it was an unexpected and striking result that full-length YscX impedes YscVc nonamerization, while YscX50 promotes it. We assume that both the stabilization of the YscVc nonamer by the YscX C-terminus and its destabilization by the YscX N-terminus may only influence the dynamics of the YscV cytosolic domain without affecting nonamer formation of full-length YscV in the membrane. Because movements of subdomains within the SctV cytosolic domain have been associated with substrate-specificity switches (Matthews-Palmer, J. Struct. Biol. 2021, doi 10.1016/j.jsb.2021.107729), changes in the dynamics of YscVc by binding or dissociation of YscX may contribute to these switches.

To better reflect these points, we changed the heading of this section to “The termini of YscX oppositely influence nonamerization of YscVc”. We also changed the introductory sentences to this paragraph to provide a clearer rationale for our gel filtration experiments (lines 260-268 in version with changes tracked).

6) Fig. 8: Consider re-coloring YscG and YscY. Pink and red are hard to distinguish. The figures would be easier to read if all proteins would be labelled.

We thank Reviewer #2 for this helpful comment. We now show YscG and AcrH in a different color that should better distinguish them from YscY. Additionally, we labelled all proteins.

7) The nomenclature should be harmonized throughout the manuscript e.g. the author switch in the middle of the introduction from using the Sct nomenclature to the Yersinia protein names of YscX and YscY.

We thank Reviewer #2 for pointing this out. We now consistently use the species-specific names unless referring to protein families or proteins from more than one species. In these cases, we use the unified Sct nomenclature instead. We did not completely switch to the Sct nomenclature, as it does not cover chaperones like YscE, YscG or SycD and their homologues, which we mention frequently. Therefore, one would be left with a mixture of Sct and species-specific nomenclature, even if one wanted to only use the unified Sct nomenclature. Most of our changes are located in the abstract and introduction with a few more in the discussion.

8) Line 24: vT3SS – what does the v stands for? I assume virulence associated, but it should be clarified in the text.

Reviewer #2 is right. We now added the explanation. (line 24 in version with changes tracked)

9) Line 24/25: T3SSs are not a mechanism, they are the machineries executing a mechanism. I would advise to rephrase the first sentence.

We changed “mechanism” to “device”. (line 25 in version with changes tracked)

10) Line 28: fT3SS abbreviation not used correctly. A flagellar type III secretion system is not the same as but rather a part of the bacterial flagellum.

We thank Reviewer #2 for pointing this out. We rephrased the sentence. The sentence now reads:

“Injectisomes share high structural similarity and likely a common ancestor with the secretion system found in bacterial flagella (fT3SS).” (lines 28-29 in version with changes tracked)

11) Line 47/48: “even though significant binding occurs also on the convex side of the chaperone” – might be a clearer phrasing

We thank Reviewer #2 for pointing out that this sentence is not clear. We completely rephrased it and we hope that this made it clearer. The sentence now reads:

“The tandem arrangement of TPRs generates a convex outer side that forms extensive interactions with the translocator and a concave binding groove where a short stretch the substrate molecule binds in an extended conformation.” (lines 46-48 in version with changes tracked)

12) Line 48: typo; correct to "gatekeeper"

We corrected this as suggested. (line 49 in version with changes tracked)

13) Line 86: virulence associated T3SS might be a better phrasing than "pathogenic T3SS"

We agree. We changed this accordingly. (line 88-89 in version with changes tracked)

14) Line 243: why not use the abbreviation T3SC?

This is a good suggestion. We changed this accordingly. (line 121 in version with changes tracked)

15) Line 260: typo; there is one "the last" too many

We removed one "the last".(line 138 in version with changes tracked)

16) Line 306-307: The authors should decide between the three or one letter amino acid code

We now use one letter code throughout.

17) Line 353: Clarify which amino acids are residues 47 and 48

We added the one letter code (Y47 and P48). (line 233 in version with changes tracked)

18) Line 357: C-terminal instead of last would be a better phrasing

We agree. We changed this accordingly. (line 237 in version with changes tracked)

19) Line 379: typo: "C-terminus" instead of "C-Terminus"

We corrected this. (line 267 in version with changes tracked)

20) Line 469: typo: "yscX deletion"

We corrected this and replaced yscx by yscX. (line 356 in version with changes tracked)

21) Fig. 6 legend, Line 699: Typo: change "bei" to "by"

We corrected this as suggested. (line 747 in version with changes tracked)

Reviewer #3 (Remarks to the Author):

The manuscript by Gilzer et al. provides the first structural information for the SctX-SctY (YscX-YscY) family of proteins that are unique, but essential components, to all of the members of the Ysc family of T3SSs. YscY is a TPR-type chaperone that directly interacts with the secreted YscX protein. The investigators also present initial structural information on the interaction of the YscX-YscY complex with the SctV (YscV) component of the export apparatus. These structures suggest that the YscX-YscY complex interacts with YscV in two ways: (1) a conserved interaction between the N-terminal region of YscX and YscY with an SD2-SD4 cleft that is similar in location to the region of SctV (FlhA) involved in the interaction with flagellar substrate-chaperone complexes; and (2) what appears to be a unique interaction between the essential YscX C-terminal region and the SD4 domains of adjacent YscV protomers. The conserved interaction likely represents an interaction involved in targeting YscX for secretion; whereas, the unique interaction likely points to the unique function of YscX (and YscY) in the T3S process. Functional studies (pull downs and chromatography) confirm the YscX-YscY interaction, and role of the YscX C-terminal region, with YscV. Furthermore, chromatography studies suggest the N-terminal region of YscX plays an important role in regulating the multimerization of the YscV nanomer. Overall, the structural and functional results presented hash well with previous results presented on the function of YscX, YscY and YscV in the T3S process and this information is well discussed in relation to the structural information provided. Overall, the manuscript presents novel and noteworthy results that will be of significant interest to the T3SS and flagellar fields of study.

The results presented support the conclusions presented by the investigators. Importantly, the results and methods provide information that will facilitate further investigations into the mechanism by which YscX-YscY and YscV contribute to the T3S process.

We thank Reviewer #3 for his time and for the positive assessment of our work.

Additional comments:

1. Structural comparisons (Fig. 8) suggest that YscY is a class 3 chaperone; however, this is not stated.

We did not state this, as we do not think that it is strictly true. Chaperone classes have typically been defined operationally by the function of their substrates and not based on homology or structural similarity. According to this classification, class I chaperones bind to effectors and class II to hydrophobic translocators. The definition of class III is a bit fuzzier and appears to comprise a structurally heterogeneous group of proteins that bind flagellin or the needle protein. We list some of these definitions below. The function of YscX is unknown and it appears to be neither effector, nor translocator nor needle component. Hence, it is hardly possible to fit its chaperone YscY into the classification scheme, which is based on the function of the substrate. Therefore, we really would prefer to not call YscY a class III chaperone. Instead we now explicitly mention “structural similarity to class III chaperones” in line 334 (in version with changes tracked).

As far as we know, introduction of class III goes back to 2003 (Parsot et al., Curr Opin Microbiol. doi: 10.1016/s1369-5274(02)00002-4.) The authors wrote “Chaperones of the flagellar export system, which is proposed to be the ancestor of TT3 systems, may constitute another class (class III) that is presented separately.”

In 2006 Cornelis defined class III as proteins that assist “subunits of substructures polymerizing outside the cytosol of the bacterium”. (Nat. Rev. Microbiol. doi:10.1038/nrmicro1526). In 2016, Cornelis (Nat. Rev. Microbiol. , doi 10.1038/nrmicro1526) mentioned as an example of a class III chaperone FliS that binds flagellin (FliC). Structurally, FliS is completely unrelated to YscG.

In 2011 Fattori et al. (Protein Pept Lett., doi: 10.2174/092986611794475048) wrote, “Class III chaperones bind to the extracellular filament proteins (or flagellin rod in the orthologous flagellar system) that polymerize into a helical structure following secretion from the bacterial cell.” As an example, they mention “the CsaA chaperone in the enteropathogenic *E. coli* that binds the EspA filament protein”. Structurally, CsaA is unrelated to YscG or YscY.

Dessen and co-workers (Izore et al. Structure 2011, doi 10.1016/j.str.2011.03.015) used the term class III for PscG and YscG, but also used an operational definition and wrote, “class III chaperones sequester the needle-forming proteins, impairing self-polymerization”.

In 2016 Notti and Stebbins wrote “class III chaperones are heterodimeric TPR proteins that bind SctF needle filament protomers.” (Microbiol Spectrum, doi:10.1128/microbiolspec.VMBF-0004-2015).

2. Line 260-261. the last the last...

We deleted on “the last”. (line 138 in version with changes tracked)

3. Fig. 6b. Line 368-371. The authors suggest the large peak seen in the ternary complex might represent a 1:1:1 ternary complex, which is possible, but was a 1:1:2 complex considered as YscX-YscY interacts with 2 adjacent YscV monomers?

We did consider the possibility, but found the apparent molecular weights obtained from the gel filtration to be contradictory to such a 1:1:2 complex. We now include supplementary table S3, which lists the elution volume for all relevant peaks in our runs, their molecular mass calculated from calibration with standard proteins and what species we assigned based on it.

A 1:1:2 complex with a molecular mass of 105.8 kDa would be too large to explain the peak at 14.93 mL (apparent M_r =57.7 kDa), which most likely corresponds to YscVc alone, as the elution volume is identical to the run of YscVc alone.

For the peak at 13.72 mL (apparent M_r = 104.6 kDa), a 1:1:2 complex appears possible. However, we do not feel confident to assign a definitive stoichiometry to any peak involving both YscV and YscXY due to (a) the already dynamic assembly of YscV, and (b) the comparatively slight difference in mass that the binding of a single YscXY causes to any larger complex.

REVIEWERS' COMMENTS

Reviewer #1 (Remarks to the Author):

I have no concerns with the current version of Gilzer et al. being published immediately in Nature Communications. My concerns regarding the resolution and the results of the L test have been very carefully and thoughtfully addressed. My more minor concerns regarding the text, the rmsd's, labeling of figures, and the role of the transmembrane region in oligomerization have been addressed. In particular, the discussion section seems to better describe the authors results. I think these minor changes strengthen the paper.

Best regards,

Ben Spiller

Reviewer #2 (Remarks to the Author):

Gilzer et al. substantially revised their manuscript and appropriately addressed my concerns. I recommend publication of the manuscript.

Reviewer #3 (Remarks to the Author):

The authors have responded satisfactorily to all of my previous comments and appear to have addressed the other reviewers comments: although, I do not have the structural biology expertise to evaluate many of these comments/responses. Overall, I feel the manuscript is novel and makes a significant contribution to the T3SS field.

Reviewer #1 (Remarks to the Author):

I have no concerns with the current version of Gilzer et al. being published immediately in Nature Communications. My concerns regarding the resolution and the results of the L test have been very carefully and thoughtfully addressed. My more minor concerns regarding the text, the rmsd's, labeling of figures, and the role of the transmembrane region in oligomerization have been addressed. In particular, the discussion section seems to better describe the authors results. I think these minor changes strengthen the paper.

Best regards,

Ben Spiller

We thank Dr. Spiller for reviewing the revised version of our manuscript and his positive assessment of our work.

Reviewer #2 (Remarks to the Author):

Gilzer et al. substantially revised their manuscript and appropriately addressed my concerns. I recommend publication of the manuscript.

We thank Reviewer #2 for reviewing the revised version of our manuscript and for the positive assessment of our work.

Reviewer #3 (Remarks to the Author):

The authors have responded satisfactorily to all of my previous comments and appear to have addressed the other reviewers comments: although, I do not have the structural biology expertise to evaluate many of these comments/responses. Overall, I feel the manuscript is novel and makes a significant contribution to the T3SS field.

We thank Reviewer #3 for reviewing the revised version of our manuscript and for the positive assessment of our work.